# Effects of using immersive virtual reality on time and steps during a locomotor task in young adults

Alexandre Renaux[1,2], Frédéric Muhla[2,3], Fabien Clanché[3], Philippe Meyer[4], Séverine Maïaux[4], Sophie Colnat-Coulbois[1], Gauchard Gérome[1,2,3]*

1 EA 3450 DevAH, Development, Adaptation and Handicap, Faculty of Medicine, Université de Lorraine, CS 50184, Vandœuvre-lès-Nancy, France, 2 CARE Grand Est, Research and Expertise Support Center, Nancy, France, 3 UFR STAPS, Faculty of Sport Science, Université de Lorraine, CS 30156, Villers-lès- Nancy, France, 4 OHS Centre Florentin, Nancy, France

* gerome.gauchard@univ-lorraine.fr

## Abstract

Immersive virtual reality makes possible to perceive and interact in a standardized, reproductible and digital environment, with a wide range of simulated situations possibilities. This study aimed to measure the potential effect of virtual reality on time and number of steps when performing a locomotor task, in a young adult's population. Sixty young adults (32W, 28M, mean age 21.55 ± 1.32), who had their first immersive virtual reality experience, performed a locomotor task based on "Timed Up and Go" (TUG) task in real, in virtual reality in a stopped train and in virtual reality in a moving train. Time and number of steps variables representing primary locomotion indicators were measured and compared between each condition. Results showed significant increases in time and number of steps in the two virtual reality conditions compared to real but not between the two virtual reality conditions. There was an effect of virtual reality in young adults when performing the locomotor task. It means that technological and digital characteristics of the immersive virtual reality experience led to modify motor strategies employed. Adding a plausible visual optic flow did not appear to affect motor control further when the information is negligible and not essential for performing the task.

## Introduction

In recent years, virtual reality technologies have been considerably developed and became affordable [1]. Important investments were engaged for improving devices. Immersive virtual reality (IVR) principle consists in immersing a person in a surrounding virtual environment. Virtual reality requires at least a 360-degree visual environment projection to be considered immersive, which is possible using a Head Mounted Display (HMD). That allows being immersed in a digitally created environment in which the user can perceive, act and interact [2]. Users can thus move and act according to appropriate sensory interfaces. This is currently mainly visual and auditory information and feedback that are correlated with the motor

**Funding:** The author(s) received no specific funding for this work.

**Competing interests:** The authors have declared that no competing interests exist.

actions performed, but this is likely to evolve thanks to innovations for multimodal immersion, haptic feedbacks or even olfactory information. The environments, on the other hand, can be fantastic or realistic, with endless possibilities of scenarios conception.

Individuals seem to use familiar sensorimotor patterns to evolve in an immersive virtual environment, which are built up on the mental organization and the automation of actions acquired during childhood [3]. For example, walking in a virtual environment seems carried out automatically as in real life [2]. The ability to produce and to script a fully customizable and modifiable virtual environments makes IVR very interesting. The situations are thus reproducible, repeatable and can simulate difficult situations to achieve in reality, with safety [4]. IVR experienced its development in video games but is now used in training, education, engineering and especially in public health. Many benefits on functional capacities were noticed in rehabilitation centers [5], due to a strong motivation and persistent engagement of people who participate in these programs [6]. A virtual reality scenario can be enriched with multiple cognitive and motor tasks [7] as well as visual perceptual information [8], having an influence on sensorimotor integration [9]. It could make possible to use IVR as a tool for evaluating locomotion, balance, dual task by simulating daily basis situations. This study is a part of the line of research conducted by our laboratory which aims precisely to investigate the benefits of IVR use as a new method to assess individual's functional abilities or to predict risk of falling.

To pursue this overall objective, it was firstly necessary to know if it did exist an IVR effect when performing the same task in real and in IVR conditions. Many studies have been carried out to measure the consequences of a virtual situation compared to a real situation. The use of virtual reality seems to be relevant in studying, evaluating or training locomotor tasks, as visuomotor behaviour is essentially similar between real and virtual [10–12]. Gait parameters can nevertheless be modified, in particular with an increase in the number of steps, with an increase in variability in stride velocity and step width [13], resulting in a decrease in gait speed [10, 14]. In addition, it has been shown that dynamic postural control in adults can be affected by IVR compared to a real situation; increases center of pressure displacement and increases in motor response time have been recorded [15, 16]. These results were consistent with two studies conducted by our research team, showing that an IVR effect was measured on elderly with a significant increase in time and number of steps parameters when performing a "Timed Up and Go" (TUG) locomotor task [17, 18].

The TUG is initially a reference test for assessing functional mobility in the elderly, consisting in performing a back and forth of three meters with seated start/end [19]. The different phases composing this locomotor task are daily activity motor tasks and therefore relevant to assess. We aim to replicate this locomotor task from a motor point of view and not such as a risk of falling predictor test, with the benefits of virtual reality: reproducibility, adaptability, and especially contextualization of the environment. Indeed, this task was carried out in a virtual environment representing a stopped train with stable visual information, and in a moving train with enriched information. These are contextualized environments simulating a daily life scenario. Moving train consisted in including a partial linear optical flow in the environment, since optic flow could influence balance and gait when moving [20]. Our study conducted with elderly showed that optical flow did not further modify motor control in IVR probably due to the characteristics of the locomotor task: locomotion distance too short and presence of a stable visual support [18].

Since previous studies of our research have only assessed elderly population, the causes of motor changes in IVR measured could therefore be linked to the technological characteristics limits of the virtual reality experience which can cause discomfort: graphics quality and latency, reduction of the field of view or even the weight of the HMD [21]; but on the other

hand, it could also intuitively be linked to the aging process in older people. The decline in visual acuity in particular [22] could have consequences for the less efficient decoding of visual cues present in the virtual environment. Moreover fear of falling is very common among the elderly and affects their daily activity [23, 24]. Performing a new and unknown task in a virtual environment could have potentially impacted their psychomotor behaviour.

A study with young adults was also needed to assess and compare the influence of immersive virtual reality. Indeed, to better understand the cause of these IVR effects, it was interesting to reproduce the similar task with a population of young adults, who have fully developed neurophysiological and psychomotor capacities. Most of their neural circuits responsible for nerve communication in the body are refined and stabilised. A process of selective stabilisation leads to a specialisation of neurons at this age [25]. The human prefrontal cortex maturation keeps going until the third decade of our existence [26]. Sensory sensors finely perceive information in the environment, which is processed very quickly, with a reaction time that is optimal at this age. It allows young people to have executive and motor functions efficient for performing tasks [27, 28].

Finally, as IVR is now offered in several fields for its advantages, especially in clinical assessing and rehabilitation, it seemed important to know more precisely the consequences of its use on motor control with a young population, not affected by ageing process and therefore relevant to better understand IVR effect during a functional and contextualized task. If aging process alone justified motor changes measured in our previous studies, then no difference between real and IVR conditions should be observed with young adults. On the other hand, if the modulations were mainly due to the characteristics of the virtual experience, then we should find an alteration in motor performance in IVR conditions. Finally, no difference between moving train and stopped train conditions should be recorded if the results with elderly were effectively related to the characteristics of the task. The aim of this study was therefore to assess the effect of immersive virtual reality on primary indicators of locomotion, time, and number of steps, when performing a locomotor task in a population of healthy young adults.

## Materials & methods

### Participants

Sixty healthy young adults aged 18 to 25 (32W, 28M, mean age 21.55 ± 1.32) volunteered to participate. They came from different faculties of the University of Lorraine and presented no neurological or musculoskeletal disorders that could affect their locomotion. They never had an IVR experience with HMD before. This study was approved by an ethical committee: CPP EST-III, N˚ID-RCB: 2018-A02637-48. Participants gave their oral consent to participate.

### Experimental conditions

Participants were asked to perform a locomotor task, based on the "Timed Up and Go" task from test by Podsiadlo & Richardson, 1991 [19]. The methodology of testing was the same than previous studies of our research project. The whole task was a set of motor actions relevant to assess daily activity: walking forward, turning around, sitting and getting up from a chair. It was feasible with our IVR device for which space was limited to 4 meters by 4 meters with outside-in tracking and without being forced to use a treadmill. It was easily contextualized in a fully reproducible visual environment. During this task, subjects were seated on a chair height approximatively 46cm with two armrests, back against to backrest, front-arms on the armrests and feet to the ground. At the word "go" they should stand up, walk three meters, turn around and return to their initial position. The oral instruction given to the participants

was "you must perform the task by walking as quickly as possible without running or putting yourself in danger". It was a locomotor task simple, fast, and easy to implement in IVR in a limited space.

Participants performed three different conditions based on this task, in a randomised controlled order to avoid any adaptation or learning effect:

- three trials in real condition (Real).

- three trials with virtual reality, VR condition in a train stopped (VR).

- three with virtual reality in motion, VR condition in a moving train (VRm).

One or more familiarisation tests were offered before each condition to ensure proper instructions understanding and compliance with the protocol.

### Immersive virtual reality task set up

The virtual application was developed on Unity game engine. The train scenarios (Fig 1) were broadcasted using a virtual reality device, an HTC Vive HMD (frame rate 90Hz, 2160 * 1200 pixels with a 110˚ field of view) and a computer with a "Nvidia Geforce GTX 1070 GPU" to allow IVR software to run smoothly. A real chair was placed in the same place as the seat present in the virtual train. A suitcase demarcated the 3 meters line, allowing to realize turn around just in front of it rather than around. All proportions of the virtual elements were consistent with a real environment, everything was on a human scale. The use of a suitcase in IVR was necessary because the HMD field of vision was more limited than the real field of vision, particularly in height. Thus, it made possible to realize the Turn-Around without leaning forward, and to avoid creating a bias. The environment was seen in first-person, visual feedbacks were consistent with the person's movement performed. Outside-in tracking provided very reliable feedback. However, IVR device was used in its lighter form so, there was no representation of the body. Participants had therefore no visual feedback from their body segments.

In the first scenario "VR", the train was stopped during the execution of the locomotor task. This condition made possible to compare the real task and the one in virtual environment with fixed visual cues. In the other one "VRm", the train was moving, causing the landscape to

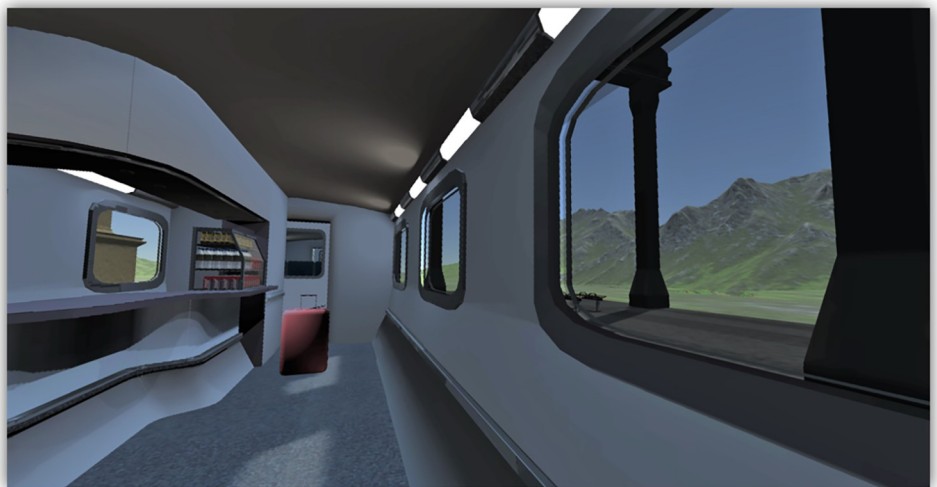

**Fig 1. Train scenario in virtual reality.** View from the chair: The task starting point.

scroll through the windows and hearing the train sounds. This condition was enriched, with a constant speed optical flow in peripheral vision, allowing to have a reproduction of a real train environment more relevant. Optic flow was in the same direction as the subject during Go phase and in the opposite direction during Return phase. The train scenario made therefore possible to preserve the different phases of the reference locomotor task and to carry it out in a visual and sound environment contextualized to a daily basis situation. Safety was ensured by a permanent presence of an experimenter to prevent any fall.

## Variables measured

Two variables were measured to characterize motor control under the different conditions, by using video (filmed with a GoPro 1920*1080p, 60fps, time was displayed on the video player MPC-HC with a 0.001s precision), with a manual post-experimentation video analysis. The temporal variable "time" and the motor variable "number of steps" were measured. Time was the main performance indicator and number of steps represented locomotion strategy modifications. Furthermore, with the fixed distance of three meters, time and number of steps were the parameters on which the gait speed and the step length depended. For instance, increasing the number of steps reflected a safer strategy adopted, witnessing a decrease in step length. Finally, the placement of the camera constrained by the space and placed on the side, slightly set back from the chair to see the back getting off the backrest and to see the participant turning around three meters away [18], only allowed to measure time and number of steps reliably and accurately.

The whole locomotor task and five specific and consecutive phases were analysed: 1) Get Up (GU) which began when the back took off from the chair, the timer was started at this moment; 2) Go (Go) which began when the participant was standing and when the first step took off to initiate the walk; 3) Turn around (TA) which started when the subject began to initiate the rotation, more precisely when the swing foot overlapped the stance foot, to touch the ground, pivoted, to initiate the rotation; 4) Return (Re) began when the swing foot overlapped the stance foot, pivoted in direction of the chair before touching the ground; and 5) Sit Down (SD) which triggered when the subject was about to turn to sit, namely when the swing foot overlapped the stance foot, to touch the ground, pivoted, to initiate the rotation before sitting down on the chair. The Sit-Down phase cutting grouping both turn-around and sit-down actions used the method of our previous studies and was explained by the strategies employed: either complete turn-around in an erected posture then sitting, or a turn-around while lowering the center of gravity and starting sitting action, this last one did not make possible to dissociate the turn-around from the sit. The timer was stopped when the participant got back to the initial sitting position. Total time and total number of steps were measured between the two contacts of the back with the backrest. This cutting phase allowed steps counts easier because time cuts were always done between two steps. Each counted step was a weight transfer from one foot to the other one, even if trampling.

## Statistical analyses

The three trials for each condition were averaged for each subject in terms of completion time and number of steps using MATLAB software. Statistics of the three trials mean values have been processed on STATISTICA. Shapiro-Wilk normality test and a sphericity test showed a non-normal distribution. Intra-individual comparisons were made between each condition to assess differences. A nonparametric Friedman's repeated measurement test for paired samples was applied. In the case of significant difference with a $p < 0.05$, a post hoc Durbin Conover test was added.

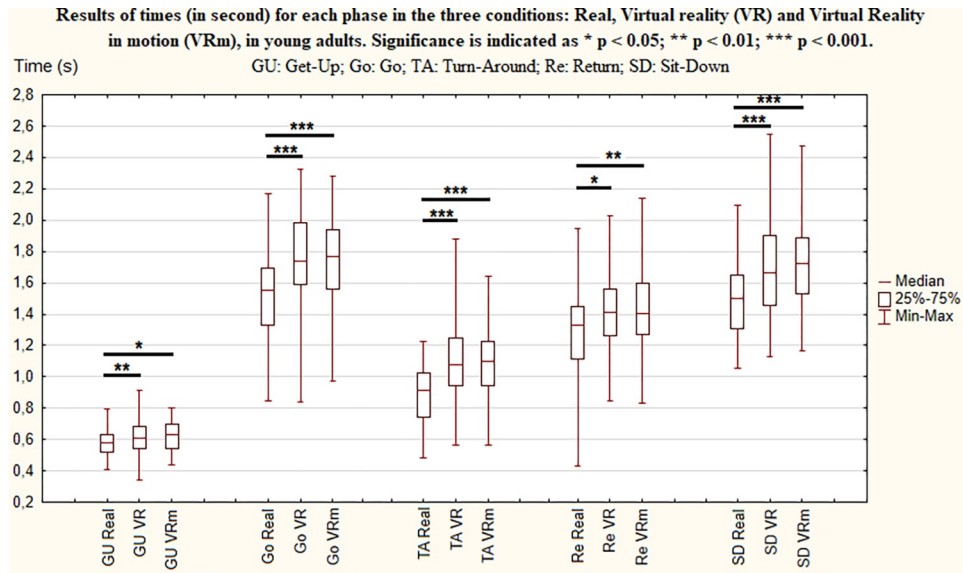

**Fig 2. Results of times (in second) for each phase in the three conditions: Real, Virtual Reality (VR) and Virtual Reality in motion (VRm), in young adults.** Significance is indicated as * $p < 0.05$; ** $p < 0.01$; *** $p < 0.001$.

## Results

### Times

Concerning Time parameter, statistically significant differences were observed between the three conditions for the whole locomotor task ($\chi^2 = 70.9$, $p < 0.001$) as well as all phases (GU: $\chi^2 = 0.23$, $p < 0.01$; Go: $\chi^2 = 46.1$, $p < 0.001$; TA: $\chi^2 = 30.6$, $p < 0.001$; Re: $\chi^2 = 7.63$, $p < 0.05$; SD: $\chi^2 = 38.2$, $p < 0.001$).

In detail, between "Real" and "VR" conditions and between "Real" and "VRm" conditions, there were significant increases in the time needed to perform the complete locomotor task and all phases. On the contrary, between "VR" and "VRm" conditions, no significant difference was observed (Figs 2 and 3).

### Steps

Concerning the number of steps, statistically significant differences were observed between the three conditions for the whole locomotor task ($\chi^2 = 65.9$, $p < 0.001$) as well as all phases (Go: $\chi^2 = 58.7$, $p < 0.001$; TA: $\chi^2 = 13$, $p < 0.001$; Re: $\chi^2 = 26.8$, $p < 0.05$), except the sitting phase (SD: $\chi^2 = 4.83$, $p = 0.09$).

In detail, between "Real" and "VR" conditions and between "Real" and "VRm", there were significant increases in the number of steps used to perform the complete locomotor task and all phases. Once again, between "VR" and "VRm" conditions, no significant difference was observed (Figs 4 and 5).

## Discussion

The aim of this study was to compare the evolution of time and number of steps used while performing a locomotor task, under three conditions with and without virtual reality for a healthy young adults' sample who had their first immersive virtual reality experience.

Many studies showed a relationship between locomotion in virtual reality and gait instability. Studies using treadmill showed that virtual environment induced gait instability in healthy

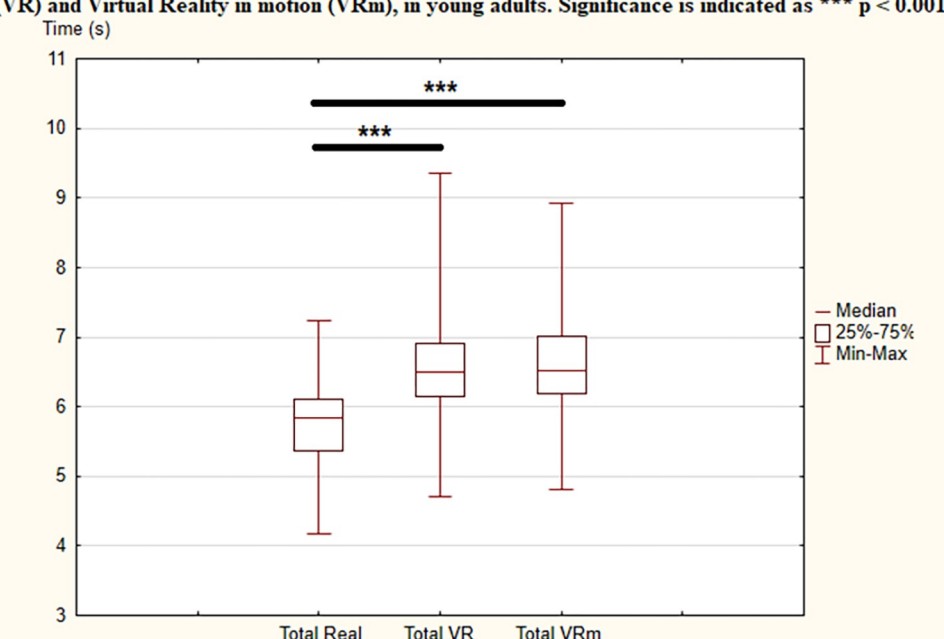

**Fig 3. Results of times (in second) for the whole locomotor task in the three conditions: Real, Virtual Reality (VR) and Virtual Reality in motion (VRm), in young adults.** Significance is indicated as *** $p < 0.001$.

subjects [29]. A recent study showed that natural locomotion was less affected even if there was a significant increase in the number of steps and cadence when walking in IVR environment [30]. In the current study, the addition of virtual reality in the locomotor task ("VR" and "VRm" conditions versus "real" condition) had an influence on motor control of young adults resulting in a significant increase in number of steps and time needed to complete the whole

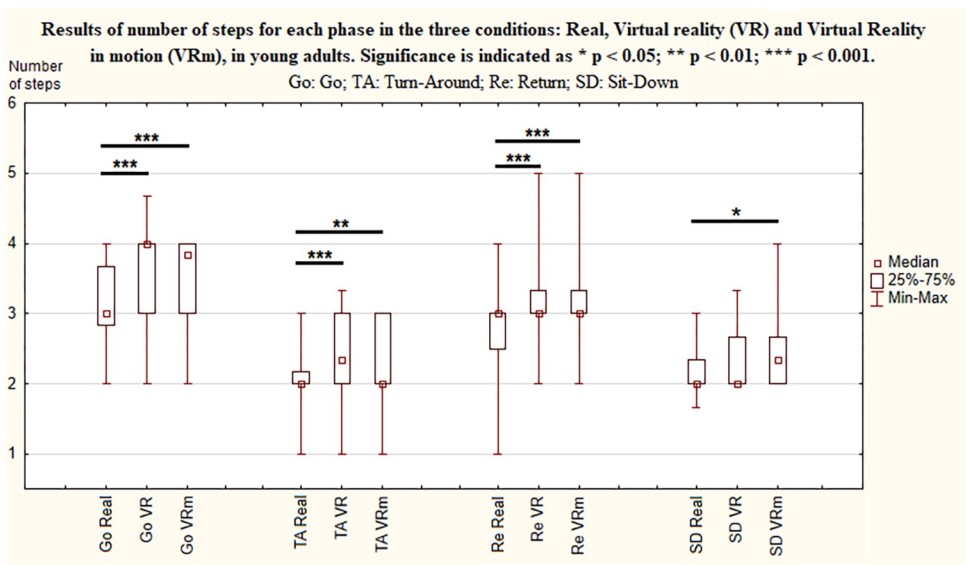

**Fig 4. Results of number of steps for each phase in the three conditions: Real, Virtual Reality (VR) and Virtual Reality in motion (VRm), in young adults.** Significance is indicated as * $p < 0.05$; ** $p < 0.01$; *** $p < 0.001$.

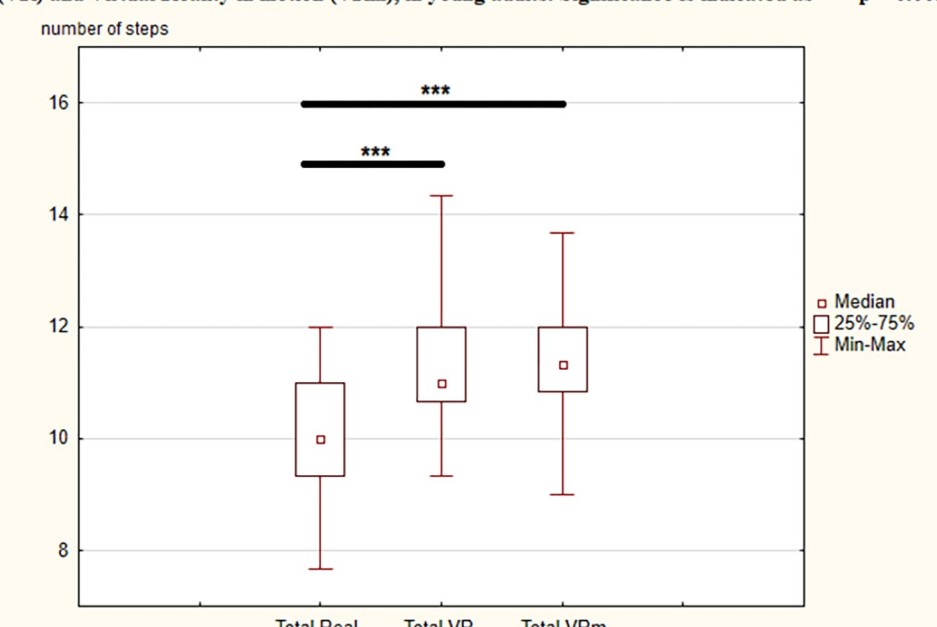

**Fig 5. Results of number of steps for the whole locomotor task in the three conditions: Real, Virtual Reality (VR) and Virtual Reality in motion (VRm), in young adults.** Significance is indicated as *** $p < 0.001$.

task in virtual conditions compared to the whole task in "real" condition; moreover, all phases were affected by a significant increase in these two parameters. These results were consistent and similar with our previous studies in older adults, which showed a significant increase in the number of steps and time taken to perform the same IVR locomotor task.

Turn-Around and Sit-Down phases required more demanding postural control with a change of orientation in space, unlike Get-Up, Go and Return phases which were only carried out on the antero-posterior axis [17]. In details, there were two strategies to perform the Turn-Around: the pivot strategy consisted of transferring the body weight to the stance foot to rotate the trunk at 180° and to redirect the swing foot left behind towards the return phase, that required two weight transfers from one foot to the other one. On the other hand, the multiple steps strategy consisted of performing this same 180° body rotation by using several steps, at least three [18]. In this study, young adults resorted to these two strategies whereas in our previous studies, elderly always needed multiple steps. More precisely, young adults used the pivot strategy almost exclusively in real condition whereas in both IVR conditions, a part of young adults have modified their strategy, from the pivot strategy to the multiple steps strategy to perform the Turn Around. Concerning the Sit-Down phase, young adults have mostly used the second strategy consisting of turning-around while lowering the center of gravity and starting sitting action in all conditions but needed more time to ensure controlled sitting in IVR conditions. To conclude, it means that young adults have modified their motor control during each phase, regardless of complexity and sensory solicitation of the phase. Time and number of steps indeed increased whether it was a simple task such as Go and Return phases, or a more complex task such as Turn-Around and Sit-Down. These results were once again comparable with the ones measured with an elderly population [18].

As young adults had efficient psychomotor skills and several memorized pattern [31], these modifications could not be linked to a desire to adopt a secure gait resulting from a chronic

fear of falling. It could be linked to the virtual environment simulating a train, a visually coherent environment, but with a few limits. The use of this new technology created a new and unfamiliar environmental context, which could create the sensation being isolated from the real world. Visual cues were relevant for informing the user about their position in the virtual environment. Indeed, technology used in this study was based on outside-in tracking, allowing precise detection of HDM orientation and position in space. It allowed precise, faithful and constant monitoring of individual's movements in real time, thus limiting the risk of motion sickness or vestibulo-ocular conflict [32]. However, the HMD only enabled visual and auditory feedback, there was so a lack of proprioceptive and haptic stimulation during virtual reality experience. Young adults could not see their own body since no virtual embodiment over an avatar was assigned, which may have affected their experience. The visual exproprioceptive feedback of body segments during movement is normally used [33]. Having a coordinated and collocated body representation with the real movements carried out in virtual environment allows to have the illusion of its own body [34] and therefore improves gesture precision and the feeling of "being present" [35]. In addition, users were aware that the elements of the virtual environment did not physically exist. This could have a negative psychological influence on the ease of performing the task. As participants have never had an IVR experience before, it is possible that cognitive and emotional reactions were modulated by this new environmental context. It could induce changes in locomotor parameters, such as walking speed [36]. There could be an adaptation or a familiarisation process if the task was repeated several times, resulting in a decrease in the IVR effect. Nevertheless, the main objective of our line of research is to determine if this technology could be used as a tool for assessing functional abilities or risk of falling. Thus, people subjected to this experiment will not necessarily have the opportunity to practice IVR before. It was therefore necessary to evaluate this IVR effect during a first IVR experience.

Wearing a HMD could also have an influence, since it imposed perceptual and biomechanical constraints [37]. First, graphics quality could still be improved, and reminded that corresponded to a digitally created environment. This aspect could be felt by focusing attention on magnified and thus easily perceivable pixels. Then, HMD characteristics, such as weight on the head and visual field limitation, forced the user to increase cervical movement amplitudes for visual exploration. It had for consequence to anticipate more the change of direction in IVR than in real with an adaptation of the head motion [38] and to amplify muscular and joints constraints. Visual field limitation may have forced users to secure changes in orientation during Turn-Around and Sit-Down phases and therefore reduced anticipation. Thus, it would seem less intuitive and automatic than in a real environment. These two phases being located consecutively to the walking phases, mechanically induced an increase in the number of steps and in time during the Go and Return phases.

To sum up, motor modifications in immersive virtual reality could result from different causalities: combination of immersive experience novelty, characteristics of the device, the lack of stimulated sensory modalities and the isolation from the real world, could lead to psychological consequences influencing the motor strategies employed. Gradual and regular exposure to virtual reality might decrease this psychological impact and lessen motor consequences. These results allowed us to conclude that aging process alone did not justify motor changes measured in our previous studies with elderly, but that modifications were mainly due to the characteristics of the virtual experience.

The experimental protocol was composed of two conditions in virtual reality which had an influence on the motor control compared to the real condition. The "VR" condition consisted of having a fixed and contextualized visual environment while the "VRm" condition consisted of having a dynamic visual and sound environment to simulate a moving train. There was a

linear optical flow, which is a typical pattern of visual motion generated at the eye when the individual is moving through the environment [39], with constant speed in VRm, in the same direction as the subject during Go phase and in the opposite direction during Return phase. Several studies have highlighted a relationship between the manipulation of visual information scrolling through the environment and walking speed, since regulation of gait involves the integration of visual, proprioceptive, and vestibular information [40]. That can result in an increase in walking speed when the flow goes forward, and a reduction walking speed when the flow goes backwards [41, 42], and more generally when the optic flow is faster or slower than walking speed, subjects modulate their locomotion inadvertently [43]. The addition of linear optical flow could therefore modify the locomotor pattern, in terms of speed, length and frequency of steps, in all individuals [44]. It was also expected to have an increase in time and number of steps in this condition compared to the train stopped, but it did not happen. Results showed indeed no significant difference between these two conditions "VR" and "VRm", in accordance with results for elderly people [18].

Several arguments could explain this result: first the characteristics of the task asked to perform, which lasted only a few seconds, with Go and Return phases of only three meters. Optic flow influence may not be noticeable over such a short distance, it does exist a minimal adaptation time needed, explained by the latency of the optic flow perception added to the latency of gait kinematic adaptation [40]. Moreover, due to its fixed nature and its proximity to the user, the train structure offered a visual support which was strategically more reliable and relevant for balance control than the landscape scrolling through the windows [18]. On the other hand, a real train moving includes vibrations and slight lateral movements. These movements classically felt by proprioceptive and vestibular sensors, were not present during the simulation and therefore potentially interfered with the virtual experience. Indeed, the proprioceptive and vestibular sensory information was actually similar to the ones in "VR" condition, and the linear optical flow was little exploited for the reasons described, which may explain that no significant difference was recorded. There is an adaptation in gait speed when proprioceptive information from body segments in motion is not congruent with visual information from optic flow [42, 43]. Finally, the task was timed, so subjects' attention was primarily focused on this constraint. In addition, the visual attention was mainly focused on the suitcase during Go phase and on the chair during Return phase rather than on the moving environment. This condition confirmed that a negligible visual enrichment of IVR scenario, results in no additional influence on motor control of young adults, in the same way than elderly [18]. It would be interesting in future studies to add multisensory stimulation to considerably enrich the virtual experience.

## Study limitations and perspectives

This study used a wired HMD to immerse individual in the virtual environment. It would be interesting to have the latest generation wireless HMD to improve the virtual experience. A study showed that better the device is, the less difference in motor control between IVR and real it is [14]. The absence of body representation over an avatar was also a limit during virtual experience because the individual had no visual feedback of his own body. In our future studies, an avatar will be included, that should allow a better precision of the movements carried out in the environment [45]. Finally, this study measured only global indicators of locomotion, the "time" and the "number of steps" when performing only one locomotor task. Next studies of our research project will focus on kinematic and kinetic determinants of gait and posture in different IVR environments. For instance, whole body kinematics data (position, velocity and acceleration of each body segment) will be analysed to determine the IVR effect when

performing complex motor tasks with interactions in the virtual environment. Locomotor and postural data recorded will then be helpful to characterize functional mobility of participants.

## Conclusion

This study demonstrated performing the locomotor task proposed in an immersive virtual reality environment led to adjust motor strategies used with a population of young adults. In the same way than elderly, it did exist an IVR effect on young's motor control. Therefore, the motor changes in IVR were essentially related to the technological and digital characteristics of virtual reality experience. Adding a plausible visual optic flow did not appear to affect motor control further when the information is negligible and not essential for performing the task. It would be very interesting in future studies to assess the magnitude of this effect in a more enriched virtual environment, both in terms of multimodal sensory stimulation and interactions to carry out.

## Supporting information

**S1 Data. Time and number of steps data.**
(XLSX)

## Author Contributions

**Conceptualization:** Alexandre Renaux, Frédéric Muhla, Sophie Colnat-Coulbois, Gauchard.

**Funding acquisition:** Alexandre Renaux, Frédéric Muhla.

**Methodology:** Alexandre Renaux, Frédéric Muhla, Fabien Clanché, Philippe Meyer, Séverine Maïaux, Sophie Colnat-Coulbois, Gauchard.

**Project administration:** Alexandre Renaux, Gauchard.

**Software:** Fabien Clanché.

**Supervision:** Sophie Colnat-Coulbois, Gauchard.

**Validation:** Gauchard.

**Writing – original draft:** Alexandre Renaux.

**Writing – review & editing:** Gauchard.

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
