## [Decision Letter · Decision Letter 0]

7 Jun 2022

PONE-D-22-05481Effects of using immersive virtual reality on time and steps during a locomotor task in young adultsPLOS ONE

Dear Dr. Renaux,

Thank you for submitting your manuscript to PLOS ONE. After careful consideration, we feel that it has merit but does not fully meet PLOS ONE’s publication criteria as it currently stands. Therefore, we invite you to submit a revised version of the manuscript that addresses the points raised during the review process. Please submit your revised manuscript by Jul 22 2022 11:59PM. If you will need more time than this to complete your revisions, please reply to this message or contact the journal office at plosone@plos.org. Please include the following items when submitting your revised manuscript:A rebuttal letter that responds to each point raised by the academic editor and reviewer(s). You should upload this letter as a separate file labeled 'Response to Reviewers'.A marked-up copy of your manuscript that highlights changes made to the original version. You should upload this as a separate file labeled 'Revised Manuscript with Track Changes'.An unmarked version of your revised paper without tracked changes. You should upload this as a separate file labeled 'Manuscript'.

We look forward to receiving your revised manuscript.

Kind regards,

Imre Cikajlo, Ph.D.

Academic Editor

PLOS ONE

Journal Requirements:

3. Please ensure that you refer to Figure 1 in your text as, if accepted, production will need this reference to link the reader to the figure.

Reviewers' comments:

Reviewer's Responses to Questions

**Comments to the Author**

1. Is the manuscript technically sound, and do the data support the conclusions?

Reviewer #1: Partly

Reviewer #2: Partly

2. Has the statistical analysis been performed appropriately and rigorously? 

Reviewer #1: Yes

Reviewer #2: I Don't Know

3. Have the authors made all data underlying the findings in their manuscript fully available?

Reviewer #1: Yes

Reviewer #2: Yes

4. Is the manuscript presented in an intelligible fashion and written in standard English?

Reviewer #1: Yes

Reviewer #2: No

5. Review Comments to the Author

Reviewer #1: The general context of this paper is the use of immersive virtual reality to study human behavior and its potential influence on this behavior. Based on a previous study where authors showed an effect of virtual reality on the kinematics of a Time Up and Go task in older adults (durations of the subtasks, more steps), authors proposed to investigate whether this effect still exists when considering young adults. To this end, authors designed an experiment where 60 young adults were asked to perform a TUG as fast as possible according to 3 blocks of 3 trials per the following conditions:

-Real World condition (Real)

-Virtual condition in a train stopped (VR)

-Virtual condition in a moving train (VRm)

Authors recorded the motion of participants using a video and computed the duration of each phase of TUG as well as the number of steps. Their results showed significant differences between Real and VR/VRm conditions (but not between VR and VRm) where the duration of TUG phases as well as the number of steps were increased in virtual conditions.

The topic of this paper is of main interest for the research community. Indeed, the understanding of how virtual reality affects motor behavior is fundamental when aiming at using this tool to analyze human motion and propose new experimental paradigms. The paper is easy to read.The statistical analysis is appropriate and relies on a high number of participants. The figures also well illustrate the results. Authors discussed the limitations of their study. However, I have 2 main concerns. My first concern is about the rationale of the study as well as the study design. Regarding the rationale of the study, the main objective of this paper is to “measure the potential effect of virtual reality on motor behavior when performing a simple locomotor task”. It would then be of main interest to better situate authors’ contribution with regards to previous work that compare participants behavior in a locomotor task in real and virtual environments (please find below some references). The literature is rich and the paper would benefit from a discussion of these past works to understand what is currently lacking or unexplored. Regarding the study design, the paper would benefit from additional justification of experimental choices. For example, why asking the participants to perform the task as fast as possible? How can the results be compared with the study of Muhla et al. with older adults where participants were asked to walk at comfort speed? Regarding the 3 conditions (Real, VR, VRm): Why considering a virtual environment representing a train? What were the characteristics of the real environment? Do they match the virtual train? If not, can it explain the difference in the kinematics? Why adding a moving train condition? What were the underlying hypotheses? For those reasons, I would argue that the paper is not ready yet for publication but a revision cycle would give the opportunity to authors to provide elements of justifications.

Please find below additional questions and comments

Introduction

lines 48-49: I would suggest to slightly rephrase this sentence, indeed, other devices such as a CAVE also exist to immerse people in virtual reality.

As discussed above, a whole area of the literature evaluating locomotor behavior in VR is missing. Please find below some examples:

FINK, P. W., P. S. FOO et W. H. WARREN (2007). « Obstacle avoidance during walking in real and virtual environments ». In : ACM Transactions on Applied Perception (TAP) 4.1, 2-es.

HOLLMAN, J. H. et al. (2007). « Does walking in a virtual environment induce unstable gait ? : An examination of vertical ground reaction forces ». In : Gait & Posture 26.2, p. 289-294.

Hollman J. H., Brey R. H., Robb R. A., Bang T. J., Kaufman K. R. (2006). Spatiotemporal gait deviations in a virtual reality environment. Gait Posture 23, 441–444.

Palmisano, C., Kullmann, P., Hanafi, I., Verrecchia, M., Latoschik, M. E., Canessa, A., ... & Isaias, I. U. (2022). A Fully-Immersive Virtual Reality Setup to Study Gait Modulation. Frontiers in Human Neuroscience, 16.

Deblock-Bellamy, A., Lamontagne, A., McFadyen, B. J., Ouellet, M. C., & Blanchette, A. K. (2021). Virtual reality‐based assessment of cognitive‐locomotor interference in healthy young adults. Journal of NeuroEngineering and Rehabilitation, 18(1), 1-10.

Berton, F., Olivier, A. H., Bruneau, J., Hoyet, L., & Pettré, J. (2019, March). Studying gaze behaviour during collision avoidance with a virtual walker: Influence of the virtual reality setup. In 2019 IEEE Conference on Virtual Reality and 3D User Interfaces (VR) (pp. 717-725). IEEE.

Agethen, P., Sekar, V. S., Gaisbauer, F., Pfeiffer, T., Otto, M., & Rukzio, E. (2018). Behavior analysis of human locomotion in the real world and virtual reality for the manufacturing industry. ACM Transactions on Applied Perception (TAP), 15(3), 1-19.

Bühler, M.A. & Lamontagne, A. (2019). Locomotor circumvention strategies in response to static pedestrians in a virtual and physical environment. Gait & Posture, 68, 201-206

Gérin-Lajoie M., Richards C. L., Fung J., McFadyen B. J. (2008). Characteristics of personal space during obstacle circumvention in physical and virtual environments. Gait Posture 27, 239–247. 10.1016/j.gaitpost.2007.03.015

Materials and Methods

Participants:

What was participants’ experience regarding virtual reality?

line 107: incomplete sentence

Experimental condition

Authors provide details about the TUG in this section, especially about the interest of considering such a task from a motor point of view. It would be interesting then to highlight what has been evaluated yet in the past works (especially locomotion, postural control) and why TUG provides new elements. Also, it would be of main interest that authors explain what can be the application of considering TUG motion in VR? What can be explored in VR using such a task that cannot be done in real conditions?

Variable measured: Can authors provide more details regarding the choice of these variables? How were the variables computed? Was it an automatic process based on kinematics (in that case, it would be relevant that authors add some details about the computation used) or was it visual annotation of the videos by the experimenters? Did authors put some markers on the participants to help identify the relevant joints? In which plane of movement the camera was located?

Results:

The results section is clearly presented.

Did authors notice some intra individual variability across the 3 repetitions of a block of condition?

Did participants report some cybersickness issues?

Discussion

Line 249: authors discussed the concept of presence. Did participants fill a presence questionnaire?

Lines 259-261: authors highlight the fact that head mounted display force the user to increase cervical amplitude, …, making the movement less natural. Did authors verify head movement amplitude in their VR conditions using HMD data?

line 261-262: visual field limitation and secure change of orientation. Do authors record the position and orientations of the helmet?

line 269-270: about the psychological impact: did authors get feedback from participants?

Lines 272-281: did participants notice that the train was moving?

Conclusion

line 301: Authors highlighted that even people borned in the digital age are affected by VR => since we do not have information about the experience of participants with VR, especially walking with a HMD, I would suggest rephrasing that sentence.

Reviewer #2: This study shows the time and number of steps that healthy young adults took to complete the Timed Up and Go test. This is a simple and short test originally designed to assess functional mobility in older adults and people with balance and gait disorders (e.g., in relation to a neurological condition). The participants conducted this task in a physical setup and in 2 virtual environments simulating a train coach. One virtual environment was static (train does not move) and the other virtual environment simulated a moving train.

The analysis of sensorimotor integration and behavioral change within virtual environments is timely and deserves much attention from research teams worldwide. However, I have some comments for the authors regarding the novelty/relevance of this study, the rationale behind the experimental design, the (lack of) comparison and contextualization based on existing literature, and the relevance and impact of the findings. More particularly, the following.

There is not a rational on why the authors decided to use the TUG in a young healthy population. Due to the simplicity of the task (and since all participants seem to have an optimal sensory and physical status), it is difficult to make any overreaching conclusions based on the results of this task. The differential results observed in the VR environments might also be assigned to the fact that participants had never experienced immersive VR before and were simply trying to complete a task within a new and unfamiliar environmental context (i.e., the virtual scenes simulating the train) while following the experimenters’ instructions.

There is no justification as to why the authors chose the variables of steps and time. For instance, steps during the Turn around and sit-down phases evaluated hardly provide any meaningful message. The manuscript needs a better justification and description for the variables chosen.

The introduction and discussions sections are seriously missing studies that already evaluated changes in motor behavior and gait performance elicited by VR environments and visual cues.

I would suggest reviewing the flow and structure of the paper, to be more objective in describing the study rationale, the research questions and hypothesis, and to focus the discussion based on the effects of virtual reality in the TUG task.

There are some statements on “motor adaptation” that, to my view, are inappropriate because the authors do not measure either motor adaptation or motor learning. If the authors believe that the statements on motor adaptation are still relevant for the study, I invite the authors to provide a definition and to further explain how adaptation is understood in this experimental study.

I would consider looking again at the statistics, particularly on how data are presented/ For instance in Table 1 the authors say data is given based on medians and interquartile range. But the numbers are very similar, and it is hard to spot or infer significant differences just by looking at the Table numbers.

The VR task needs a much better description, and explanation on why a train was chosen. In addition, what are the details of the visual scenery utilized? How the virtual environment change according to the person’s movements, is there an embodiment over an avatar, is there an avatar of the participants, what happened if the participant wearing the glasses attempts to look down to his/her feet torso, do they see a virtual body, how the different directions of the train were simulated and differentiated?

Finally, I see that the authors cite themselves several times so as third persons (e.g., ref. 10 and 20). Including sentences such as ““The methodology of testing was inspired by that of Muhla et al., 2021”. I understand if the authors prefer this style, my personal preference thought is being clear about the fact that one study builds up on previous study of the research group. Anyway, if this study makes part of a line of research, it would be interesting to understand the authors mindset on how the several studies of the research group complement each other, and perhaps more importantly, to be more explicit on the future scientific directions of the group in such line of research.

Punctual comments

Abstract

Regarding the aim of the study “This study aims to measure the potential effect of virtual reality on motor behavior when performing a simple locomotor task, in a population of young adults with fully developed functional abilities.” Also, in the manuscript appears: The aim of this study is therefore to assess the effect of immersive virtual reality on motor skills during a locomotor task in a population of healthy young adults. Do we really believe the experiment changes motor skills among participants? Also, we cannot reduce “motor behavior” to steps and time, not especially in such a short and easy task as TUG among healthy individuals. I think that the authors need to tone down the aim statement (and the claims) of the study and provide a more specific objective of the study. It of course depends on the rationale and hypothesis of the original experimental design. What was the objective? What was expected? To see if TUG in VR tasks were longer or demanding more steps in comparison to real environments?

Introduction

The opening paragraph can be improved in terms of English grammar and a reference is missing for the 30% business growth stated.

“VR principle” is not necessarily linked to a HMD (although HMDs are the most common display for VR interactions, there exist also others such as the CAVE systems). I suggest the authors to dissociate VR principles from VR tools (e.g., displays, interfaces)

The authors mention “immersion” several times and comment once on presence. As it looks to be important for the contextualization proposed in this manuscript, I suggest defining and differentiating both concepts.

Reference (3) is an EEG study comparing 2D and 3D virtual environments. I am unsure this is the best citation for taking the reader to better understand the sense of presence in VR. I suggest the authors to review previous psychological papers that describe the sense of presence in VR (e.g., papers from Prof. Mel Slater)

What are “natural sensorimotor patterns”? Are there non-natural sensorimotor patterns?

How exactly ref (2) implies that “walking in a virtual environment seems carried out unconsciously and without additional cognitive effort, as in real life”?

The second last paragraph in the introduction speaks about neural networks and neurophysiological development in young adults. But it is quite disconnected from the rest of the section. To be more enriching, I suggest the authors to explain how this information is relevant in the context of the conducted experiment; for instance, how does it affect the research questions and hypothesis of this study?

“The aim of this study is therefore to assess the effect of immersive virtual reality on motor skills during a locomotor task”. However, the introduction does not address a rich literature that evaluated VR effects of gait and locomotion (and motor behavior in general). The authors do not mention previous studies that examined similar situations, and it makes difficult to understand the novelty and the main take-away messaged from the present study. I suggest the authors to contrast the rationale and hypothesis of their study with the findings obtained from previous studies (I put out a list of studies in the bottom of my review, but there exist more papers than those in the list). In a similar vein, the Discussion should comprise a comparison to the literature ideally.

Materials & Methods

At line 107, the sentence “an immersive virtual reality device.” Seems out of context

The TUG test was originally designed to assess basic functional mobility in the elderly, and today is used also to test patients with neurological conditions, among others. The TUG is a very simple task for healthy individuals like the ones recruited in this study. It is hardly going to provoke major effects (in terms of locomotion or motor behavior) even when performed within VR environments. I am afraid that the findings of this study are limited due to this, and it makes difficult to draw conclusions about the effect of VR in “motor behavior”. I invite the authors to better justify the election of the TUG test in a healthy population, and to discuss how the simplicity of task affect the generalization of the findings and conclusions, such as the one in the abstract claiming that “There is an effect of virtual reality on the motor behavior of young adults”, or the one in the discussion claiming that “Results showed an influence for both virtual reality conditions on motor skills” (were motor skills actually affected in the individual?)

Was there any ethical procedure or approval for this study? Did the authors provide informed consent to participate?

I believe the subsection ‘virtual reality device’ describes more than only the device. I suggest renaming the subsection with a different title.

Did the participants embody an avatar that walks in the VR scenario? What happens when participants wearing the HMD look down towards their own body? Did they see a virtual body?

The manuscript needs a more expanded description and explanation of the VR scenario. For instance, the following.

• In the VRm condition, when participants raise up from the chair, did participants walk in the same direction that the train was moving? Or did they walk in the direction from which the train is coming? This is an important consideration because we have evidence that optical flow affects behavior and motor performance.

• Also, when the participants were coming back to the chair, were they able to see the chair? Did the dimensions of the chair change according to the approximation of the participant?

• How was it guarantee that participants walked the same distance (3m) and turn around at the same spot in the normal and in the VR conditions? In the VR conditions, did participants need to overpass the suitcase or just get close to it?

• If possible, add a descriptive or explanatory video to better understand the performance of the task in VR.

In the description of the Turn Around (TA) phase, the authors state “(TA) which starts when the subject begins to initiate the rotation, when actually trajectory changes to return to the chair” But these two events are not the same. One thing is “to initiate rotation” and a different thing is “trajectory changes to return to the char”. Can the authors specify where the analysis was made?

I would like to have a rationale and justification on why the authors chose time and number of steps as variables to be measured.

Did the authors assess differences between the Go and Re phases within the VR environments? I would be curious to know the comparison results, if walking in a moving train, within a VR scenario, changes when one walks in the same direction than the train vs. a direction against the train.

How was the variable “time” extracted? With a stopwatch in place during the experiment and further annotation? Or by analyzing the video in a post-experiment analysis?

Please describe the equipment used to video record the participants.

Figure 1 is not cited in the manuscript. Also, a legend is missing. Again, I believe a video would be a best option to facilitate a better understanding on how the task is performed.

Results

In Table 1, Why are there steps in the Turn Around and Sit-Down phases? Do we step when we sit down? How is it quantified?

I am hesitant about the statistical presentation of the data. The authors say values from the 3 trials were averaged. Then in Table 1, authors use Medians. Is it the median of the mean values? Also, for steps I understand the authors recorded the number of completed steps; this is, integer numbers such as 1, 2, 3, 4 or 5. Still we have data of steps such as 2,33 and 3,83. Another thing that is unclear: why the Total number is not the sum of the steps? Perhaps a better description of how data is handled will avoid confusion in the reader and serve for clarification.

Please provide better legends for all figures. Figures, together with legends, should be self-explanatory.

The Data presented in the Supplementary Material is the average from the 3 trials, correct? Can the authors provide the data of the 3 individual trials as well?

Discussion

What message do the authors want to pass with the following sentence: “have adapted and modified their motor behavior during each phase, regardless of its complexity and sensory solicitation”?

How exactly can we get to the conclusion that: “It means that young adults have adapted and modified their motor behavior during each phase, regardless of its complexity and sensory solicitation”. I do not understand why some discussion sections revolve around adaptation. Did the authors assess adaptation?

The discussion in general is lacking flow and structure. A comparison with the existing literature is seriously lacking, in particular with studies that have evaluated gait and motor changes during VR tasks.

The following sentence “Young adults could not see their own body” related to a previous comment of mine about avatar embodiment. Did the authors experience embodiment of an avatar to perform the task? Were participants assigned a virtual body?

The authors state that “As participants have never had an immersive virtual reality experience before”. Was this an inclusion criterion during participant recruitment? How the authors assessed this?

In general, I invite the authors to rethink the Discussion section, including incorporating the comments presented in the beginning of my review. In addition, a ‘study limitation’ section is missing. The authors may want to discuss what as well are the implications of the present study (e.g., scientific, clinical, technical).

The following information needs to appear in the Methods section: “In “VRm” condition, the virtual train is moving, with sounds and landscape scrolling outside the train. There is therefore a linear optical flow with constant speed, in the same direction as the subject during Go phase and in the opposite direction during Return phase.”

This is good to have a reference such as 28, but others are lacking, and the discussion about the effect of optic flow should be expanded. Also, it should be incorporated in the introduction. I also invite the authors to consider reporting on an extra analysis that compares the Go and Return phases (since optic flow is different in these phases).

In my opinion, the last 2 paragraphs of the discussion are important. First, I think they should appear as main topic of discussion. Second, references are missing. It is important that the authors review the existing literature and enrich this discussion. Below, I add a few of relevant studies.

Conclusion

Be objective in the conclusion, stick to the experiment results, and down tone overreaching claims.

General

There seems to be a lack of standardization on how the references are presented. Please review the reference format recommended by the journal.

English needs a major revision. I have spotted important flaws in the structure and flow of the sentences and connections between paragraphs. E.g., the following:

- “VR has therefore great interests in ability”. VR is a technology, it has no inner interests.

- “increased in motor response time have been recorded” > increases

- “Performed a new and unknown task”… > performing

- “allowing to realise turn around just in front of it.”

- “the train is driving,” > the train is moving

Small selection of references describing the effect of VR in locomotion and motor behavior in healthy adults

- Mohler B. J., Thompson W. B., Creem-Regehr S. H., Pick H. L., Warren W. H. (2007). Visual flow influences gait transition speed and preferred walking speed. Exp. Brain Res. 181 221–228.

- Cano Porras D, Zeilig G, Doniger GM, Bahat Y, Inzelberg R, Plotnik M. Seeing Gravity: Gait Adaptations to Visual and Physical Inclines - A Virtual Reality Study. Front Neurosci. 2020;13:1308. Published 2020 Jan 24. doi:10.3389/fnins.2019.01308

- O’Connor S. M., Donelan J. M. (2012). Fast visual prediction and slow optimization of preferred walking speed. J. Neurophysiol. 107 2549–2559

- Guerin P., Bardy B. G. (2008). Optical modulation of locomotion and energy expenditure at preferred transition speed. Exp. Brain Res. 189 393–402.

- Prokop T., Schubert M., Berger W. (1997). Visual influence on human locomotion modulation to changes in optic flow. Exp. Brain Res. 114 63–70

- Plotnik M., Azrad T., Bondi M., Bahat Y., Gimmon Y., Zeilig G., et al. (2015). Self-selected gait speed-over ground versus self-paced treadmill walking, a solution for a paradox.

- Thompson J. D., Franz J. R. (2017). Do kinematic metrics of walking balance adapt to perturbed optical flow? Human Movement Science 54 34–40.

Sometimes the authors write virtual reality in full and in other parts appears VR, the abbreviation. I suggest the authors to standardize. The same goes to Head Mounted display vs. HMD.

6. PLOS authors have the option to publish the peer review history of their article (what does this mean?). If published, this will include your full peer review and any attached files.

Reviewer #1: No

Reviewer #2: **Yes: **Cano Porras D.

---

## [Author Response · Author response to Decision Letter 0]

19 Jul 2022

Response to Reviewer #1

Global point:

Reviewer’s comments: 

The topic of this paper is of main interest for the research community. Indeed, the understanding of how virtual reality affects motor behavior is fundamental when aiming at using this tool to analyze human motion and propose new experimental paradigms. The paper is easy to read. The statistical analysis is appropriate and relies on a high number of participants. The figures also well illustrate the results. Authors discussed the limitations of their study. However, I have 2 main concerns. My first concern is about the rationale of the study as well as the study design. Regarding the rationale of the study, the main objective of this paper is to “measure the potential effect of virtual reality on motor behavior when performing a simple locomotor task”. It would then be of main interest to better situate authors’ contribution with regards to previous work that compare participants behavior in a locomotor task in real and virtual environments (please find below some references). The literature is rich and the paper would benefit from a discussion of these past works to understand what is currently lacking or unexplored. Regarding the study design, the paper would benefit from additional justification of experimental choices. 

For example, why asking the participants to perform the task as fast as possible? How can the results be compared with the study of Muhla et al. with older adults where participants were asked to walk at comfort speed? 

Authors response:

It is the same idea but explained differently. Comfortable speed meant the fastest speed without putting yourself in danger, without risk of falling, so it was a comfortable speed in elderly. In young adults, this notion of risk of falling does not exist, it was the same instruction to perform the task as quickly as possible while remaining natural, without running and ultimately in a comfortable way.

Reviewer’s comments: 

Regarding the three conditions (Real, VR, VRm): Why considering a virtual environment representing a train? What were the characteristics of the real environment? Do they match the virtual train? If not, can it explain the difference in the kinematics? Why adding a moving train condition? What were the underlying hypotheses? For those reasons, I would argue that the paper is not ready yet for publication, but a revision cycle would give the opportunity to authors to provide elements of justifications.

Authors response:

We consider immersive virtual reality (IVR) as a technology allowing the reproduction of contextualized and reproducible environments. IVR could thus be a new tool for assessing the risk of falls in the elderly. Indeed, simulating a scenario that represents a real-life environment (for instance: a train) is the advantage offered by virtual reality. 

The “Timed Up and Go” is a very recognized test to assess functional mobility in older adults, we therefore adapted the locomotor task in a virtual train. The current study as well as the studies conducted previously are preliminary phases to better understand IVR effect on motor control and age-related effect. 

Concerning the different conditions (real, VR and VRm):

• Stopped train condition (VR) makes possible to compare the task with the real test (Real) in a contextualized virtual environment with fixed visual cues. 

• Moving train condition (VRm) is considered more enriched, with an optical flow, that adds relevance to the reproduction of a real environment. The condition with optical linear flow should be more disruptive than the condition with fixed visual cues. However, no influence has been recorded with elderly people, but we wanted to evaluate it in young adults as well.

Additional points:

- Point 1: 

Reviewer’s comments: 

Introduction lines 48-49: I would suggest to slightly rephrase this sentence, indeed, other devices such as a CAVE also exist to immerse people in virtual reality.

Authors response:

The CAVE is a semi-immersive device, so we specified only immersive VR device was considered (HMD).

- Point 2:

Reviewer’s comments: 

Participants: What was participants’ experience regarding virtual reality?

Authors response:

Participants had never experienced an immersive virtual reality experience with HMD before. It was specified at the end of the participants’ paragraph.

- Point 3:

Reviewer’s comments: 

Experimental condition: Authors provide details about the TUG in this section, especially about the interest of considering such a task from a motor point of view. It would be interesting then to highlight what has been evaluated yet in the past works (especially locomotion, postural control) and why TUG provides new elements. Also, it would be of main interest that authors explain what can be the application of considering TUG motion in VR? What can be explored in VR using such a task that cannot be done in real conditions?

Authors response:

- TUG task is a set of motor actions relevant to daily activity (walking forward, turning around, sitting, and getting up from a chair), which are interesting for assessing. Previous studies performed TUG locomotor task in a virtual reality environment but were focused on elderly people. Choosing the same task allows to compare with these studies.

- IVR space is limited to 4 by 4 meters with outside-in tracking, this locomotor task can thus be carried out in this space. Moreover, that makes possible to perform natural locomotion, without having to use a treadmill like most studies. Analysis of motor control therefore seems more appropriated.

- Classic TUG task is performed in an empty room, sanitized from any stimulations, which does not correspond to everyday life. Performing this task in IVR has the advantage of being able to contextualize the situation in a fully reproducible visual and sound environment. The underlying objective is to simulate a situation of everyday life in the most ecological way as possible.

- The moving train condition allows to have a condition enriched of stimulations. Studies showed that optical linear flow can disrupt and alter motor control: we wanted to measure it. 

- Point 4:

Reviewer’s comments: 

Variable measured: Can authors provide more details regarding the choice of these variables? How were the variables computed? Was it an automatic process based on kinematics (in that case, it would be relevant that authors add some details about the computation used) or was it visual annotation of the videos by the experimenters? Did authors put some markers on the participants to help identify the relevant joints? In which plane of movement, the camera was located?

Authors response:

- We measured the same indicators (time and number of steps) as in previous studies of our research team to be able to compare. More precisely, "time" is the main evaluation indicator to highlight a modification in performing the task; and number of steps represents adjustments of locomotion (for example: adopting a safe strategy lead to increase the number of steps, by reducing the length). Finally, time and number of steps are the primary indicators defining locomotion. 

- These variables were recorded using video, with a camera placed in the corner of the room. The space in which we experimented was too narrow for the placement of cameras in the sagittal or frontal planes.

- It was not an automated process but a manual post experiment video analysis, without any marker placed on the participants. These are cutting criteria directly recorded on video: cutting points that separated each phase are explained in methodology.

- Point 5:

Reviewer’s comments: 

Results: The results section is clearly presented.

Did authors notice some intra individual variability across the 3 repetitions of a block of condition?

Authors response:

There was no high individual variability across the 3 trials for each condition. That was compared to assess a possible learning effect, which was not significant.

- Point 6:

Reviewer’s comments: 

Results: Did participants report some cybersickness issues?

Authors response:

Cybersickness was not clearly assessed by questionnaire but simply by verbal feedback: no participant mentioned problems with motion sickness. The exposure time was very short, all settings of the head-mounted display were made to allow them being in the best conditions. In addition, there was no vestibulo-ocular conflict since visual feedback were consistent with the movement performed. Finally, the scrolling speed of the landscape was constant to avoid any acceleration effect in “VRm” condition. The benefit of outside-in tracking provides very reliable feedback.

- Point 7:

Reviewer’s comments: 

Discussion Line 249: authors discussed the concept of presence. Did participants fill a presence questionnaire?

Authors response:

This concept was not specifically evaluated in this study. We removed this notion from the article to avoid any confusion.

- Point 8:

Reviewer’s comments: 

Discussion Lines 259-261: authors highlight the fact that head mounted display force the user to increase cervical amplitude, …, making the movement less natural. Did authors verify head movement amplitude in their VR conditions using HMD data?

Authors response:

No, it is just an explanatory hypothesis, in connection with the reduced field of vision (110°) in the HMD compared to normal vision reaching 270°.

- Point 9:

Reviewer’s comments: 

Discussion line 261-262: visual field limitation and secure change of orientation. Do authors record the position and orientations of the helmet?

Authors response:

Position and orientation of the head-mounted display are indeed measured automatically by the device, but there are still limitations which do not allow to analyze it. First, orientation of the HMD does not necessarily reflect the orientation of the gaze (no consideration of head movement relative to the trunk and no consideration of eyes movement). Secondly, there is no possible comparison with the real condition in the absence of HMD.

- Point 10:

Reviewer’s comments: 

Discussion line 269-270: about the psychological impact: did authors get feedback from participants?

Authors response:

We got verbal feedback. Indeed, participants were impressed by the consistency of the visual reproduction in relation to the movement performed. We can imagine they would likely feel even more comfortable and engaged with prolonged exposure. Each condition needed less than a minute, leaving little time to be psychologically fully in and detach from the real world. Prolonged exposure time would probably make possible to forget the physical world and be fully engaged in the virtual world.

- Point 11:

Reviewer’s comments: 

Discussion Lines 272-281: did participants notice that the train was moving?

Authors response:

They knew it. On the one hand, instructions given by experimentators specified that train was moving. On the other hand, visual and sound information offered by virtual environment dispelled any doubt. However, attention dedicated to the task and the structure of the train offering many fixed visual cues probably overlooked the surrounding moving environment.

- Point 12:

Reviewer’s comments: 

Conclusion line 301: Authors highlighted that even people borned in the digital age are affected by VR => since we do not have information about the experience of participants with VR, especially walking with a HMD, I would suggest rephrasing that sentence.

Authors response:

We rewrote the conclusion paragraph.  

Response to Reviewer #2

Global point:

Reviewer’s comments: 

This study shows the time and number of steps that healthy young adults took to complete the Timed Up and Go test. This is a simple and short test originally designed to assess functional mobility in older adults and people with balance and gait disorders (e.g., in relation to a neurological condition). The participants conducted this task in a physical setup and in 2 virtual environments simulating a train coach. One virtual environment was static (train does not move) and the other virtual environment simulated a moving train.

The analysis of sensorimotor integration and behavioral change within virtual environments is timely and deserves much attention from research teams worldwide. However, I have some comments for the authors regarding the novelty/relevance of this study, the rationale behind the experimental design, the (lack of) comparison and contextualization based on existing literature, and the relevance and impact of the findings. More particularly, the following.

There is not a rational on why the authors decided to use the TUG in a young healthy population. Due to the simplicity of the task (and since all participants seem to have an optimal sensory and physical status), it is difficult to make any overreaching conclusions based on the results of this task. The differential results observed in the VR environments might also be assigned to the fact that participants had never experienced immersive VR before and were simply trying to complete a task within a new and unfamiliar environmental context (i.e., the virtual scenes simulating the train) while following the experimenters’ instructions.

Authors response:

This study is part of a line of studies contextualizing locomotor tasks in an immersive virtual reality environment (a train). The aim is to determine if the use of IVR could be used as a new tool for the detection of risk of falling in the elderly. 

Preliminary studies are carried out to assess the effect of IVR use on primary indicators of locomotion: time and number of steps. We measured significant increases in time and number of steps in IVR condition compared to real with an elderly population. In this study, we reproduce the same methodology with a young population to determine the effect of IVR concerning the same variables. If aging process alone justifies motor changes measured in our previous studies, then no difference between real and IVR conditions should be observed with young adults. On the other hand, if the modulations are mainly due to the characteristics of the virtual experience, then we should find an alteration in motor performance in IVR conditions with young people. 

Finally, we do not apply TUG task as fall risk predictor test in this study but as a reference motor task. We thus examine the interest of considering such a task from a motor point of view. Indeed, TUG locomotor task is a set of relevant motor actions for assessing daily activity tasks.

Reviewer’s comments: 

There is no justification as to why the authors chose the variables of steps and time. For instance, steps during the Turn around and sit-down phases evaluated hardly provide any meaningful message. The manuscript needs a better justification and description for the variables chosen.

Authors response:

We measured the same indicators (time and number of steps) as in previous studies of our research team to be able to compare. More precisely, "time" is the main evaluation indicator to highlight a modification in performing the task; and number of steps represents adjustments of locomotion (for example: adopting a safe strategy lead to increase the number of steps, by reducing the length). Finally, time and number of steps are the primary indicators defining locomotion. 

Concerning Turn Around and sit-down phases, young people can perform their turn-around with a pivot (only single step) in real condition, whereas they needed more steps adjustments to perform turn-around in IVR condition. They less anticipated the turn-around in IVR and thus modified their motor strategy. It is interesting to measure it from this point of view.

Reviewer’s comments: 

The introduction and discussions sections are seriously missing studies that already evaluated changes in motor behavior and gait performance elicited by VR environments and visual cues.I would suggest reviewing the flow and structure of the paper, to be more objective in describing the study rationale, the research questions and hypothesis, and to focus the discussion based on the effects of virtual reality in the TUG task.

Authors response:

Thank you very much for your comments, we improved introduction and discussion accordingly. 

Reviewer’s comments: 

There are some statements on “motor adaptation” that, to my view, are inappropriate because the authors do not measure either motor adaptation or motor learning. If the authors believe that the statements on motor adaptation are still relevant for the study, I invite the authors to provide a definition and to further explain how adaptation is understood in this experimental study.

Authors response:

You are right, motor adaptation is a form of motor learning, a process of acquiring and restoring locomotor patterns. That was not evaluated in our study, the term is inappropriate. Motor modifications seem more appropriated.

Reviewer’s comments: 

I would consider looking again at the statistics, particularly on how data are presented/ For instance in Table 1 the authors say data is given based on medians and interquartile range. But the numbers are very similar, and it is hard to spot or infer significant differences just by looking at the Table numbers.

Authors response:

Table 1 displayed descriptive data only but has been removed cause graphics and boxplot are more appropriate to visualize significant differences. 

Reviewer’s comments: 

The VR task needs a much better description, and explanation on why a train was chosen. In addition, what are the details of the visual scenery utilized? How the virtual environment change according to the person’s movements, is there an embodiment over an avatar, is there an avatar of the participants, what happened if the participant wearing the glasses attempts to look down to his/her feet torso, do they see a virtual body, how the different directions of the train were simulated and differentiated?

Authors response:

The advantage of virtual reality is to be able to contextualize a reproducible scenario. It is possible to perform the locomotor task in a contextualized environment of everyday life, a train. We get up from the seat, walk 3 meters and come back to sit down. It is the same test but in a contextualized visual and sound environment.

Visual feedbacks were consistent with the person’s movement performed. The benefit of outside-in tracking provides very reliable feedback.

There was no embodiment over an avatar. Participants evolved in the environment with first person view and had no visual feedback of their own body. When they looked down, they only saw the floor of the train, not their body. That has been clarified in the methodology section.

Reviewer’s comments: 

Finally, I see that the authors cite themselves several times so as third persons (e.g., ref. 10 and 20). Including sentences such as ““The methodology of testing was inspired by that of Muhla et al., 2021”. I understand if the authors prefer this style, my personal preference thought is being clear about the fact that one study builds up on previous study of the research group. Anyway, if this study makes part of a line of research, it would be interesting to understand the authors mindset on how the several studies of the research group complement each other, and perhaps more importantly, to be more explicit on the future scientific directions of the group in such line of research.

Authors response:

Your comment is very relevant, we have therefore modified the structure of the study to allow readers better understand our line of research. 

Additional points:

- Point 1: 

Reviewer’s comments: 

Abstract: Regarding the aim of the study “This study aims to measure the potential effect of virtual reality on motor behavior when performing a simple locomotor task, in a population of young adults with fully developed functional abilities.” Also, in the manuscript appears: The aim of this study is therefore to assess the effect of immersive virtual reality on motor skills during a locomotor task in a population of healthy young adults. Do we really believe the experiment changes motor skills among participants? Also, we cannot reduce “motor behavior” to steps and time, not especially in such a short and easy task as TUG among healthy individuals. I think that the authors need to tone down the aim statement (and the claims) of the study and provide a more specific objective of the study. It of course depends on the rationale and hypothesis of the original experimental design. What was the objective? What was expected? To see if TUG in VR tasks were longer or demanding more steps in comparison to real environments?

Authors response:

Associating motor behavior with only time and number of steps is indeed too simplistic. We have therefore nuanced the terms throughout the study. The main objective is to assess the effect of IVR use on time and number of steps when performing a locomotor task. Underlying hypothesis have been added in the introduction section, in consideration with previous studies. 

- Point 2:

Reviewer’s comments: 

Introduction: The opening paragraph can be improved in terms of English grammar and a reference is missing for the 30% business growth stated. 

“VR principle” is not necessarily linked to a HMD (although HMDs are the most common display for VR interactions, there exist also others such as the CAVE systems). I suggest the authors to dissociate VR principles from VR tools (e.g., displays, interfaces).

Authors response:

This paragraph has been modified and clarified. CAVE system is a semi-immersive device, so we specified only immersive VR device was considered (HMDs). “Immersive virtual reality (IVR) principle consists in immersing a person in a surrounding virtual environment. Virtual reality requires at least a 360-degree visual environment projection to be considered immersive, which is possible using a Head Mounted Display (HMD).”

- Point 3: 

Reviewer’s comments: 

Introduction: The authors mention “immersion” several times and comment once on presence. As it looks to be important for the contextualization proposed in this manuscript, I suggest defining and differentiating both concepts. 

Reference (3) is an EEG study comparing 2D and 3D virtual environments. I am unsure this is the best citation for taking the reader to better understand the sense of presence in VR. I suggest the authors -to review previous psychological papers that describe the sense of presence in VR (e.g., papers from Prof. Mel Slater)

Authors response:

This study does not assess the feeling of presence and the level of immersion. To avoid any confusion, we have decided to remove these notions.

- Point 4:

Reviewer’s comments: 

Introduction: What are “natural sensorimotor patterns”? Are there non-natural sensorimotor patterns?

Authors response:

It means usual or familiar sensorimotor pattern; terms have been modified.

- Point 5:

Reviewer’s comments: 

Introduction: How exactly ref (2) implies that “walking in a virtual environment seems carried out unconsciously and without additional cognitive effort, as in real life”?

Authors response:

Basing on the psychological concept of scheme (Piaget, 1979): "individual uses automatisms in his sensorimotor activities he has assimilated in the real world since childhood. It is the mental organization of actions".

Fuchs, 2018 thus wrote “In an interactive virtual environment, the person uses the same approach as in the real world, to organize the virtual according to a set of spatio-temporal and causal rules known”

In this case, walking will be done unconsciously as in reality, and without additional cognitive effort

This paragraph is available in the chapter 6 "interfaçage comportemental" of "Théorie de la réalité virtuelle" by Fuchs, 2018.

- Point 6:

Reviewer’s comments: 

Introduction: The second last paragraph in the introduction speaks about neural networks and neurophysiological development in young adults. But it is quite disconnected from the rest of the section. To be more enriching, I suggest the authors to explain how this information is relevant in the context of the conducted experiment; for instance, how does it affect the research questions and hypothesis of this study?

Authors response:

This paragraph argues that young adults included in this study have optimal functional abilities linked to their maximum neuronal development at this age, in comparison with elderly population affected by ageing process.

- Point 7:

Reviewer’s comments: 

Introduction: “The aim of this study is therefore to assess the effect of immersive virtual reality on motor skills during a locomotor task”. However, the introduction does not address a rich literature that evaluated VR effects of gait and locomotion (and motor behavior in general). The authors do not mention previous studies that examined similar situations, and it makes difficult to understand the novelty and the main take-away messaged from the present study. I suggest the authors to contrast the rationale and hypothesis of their study with the findings obtained from previous studies (I put out a list of studies in the bottom of my review, but there exist more papers than those in the list). In a similar vein, the Discussion should comprise a comparison to the literature ideally.

Authors response:

Thank you very much for your comments, we improved introduction and discussion with literature that evaluated VR effects of gait and locomotion.

- Point 8:

Reviewer’s comments:

Materials & Methods: At line 107, the sentence “an immersive virtual reality device.” Seems out of context

Authors response:

It was a wrong sentence, a mistake. This sentence has been removed. 

- Point 9:

Reviewer’s comments: 

Materials & Methods: The TUG test was originally designed to assess basic functional mobility in the elderly, and today is used also to test patients with neurological conditions, among others. The TUG is a very simple task for healthy individuals like the ones recruited in this study. It is hardly going to provoke major effects (in terms of locomotion or motor behavior) even when performed within VR environments. I am afraid that the findings of this study are limited due to this, and it makes difficult to draw conclusions about the effect of VR in “motor behavior”. I invite the authors to better justify the election of the TUG test in a healthy population, and to discuss how the simplicity of task affect the generalization of the findings and conclusions, such as the one in the abstract claiming that “There is an effect of virtual reality on the motor behavior of young adults”, or the one in the discussion claiming that “Results showed an influence for both virtual reality conditions on motor skills” (were motor skills actually affected in the individual?)

Authors response:

The aim of this study is specially to evaluate the effect of IVR use on primary indicators of locomotion when performing a representative locomotor task, without high complexity or added double task. This study makes possible to confirm that it does exist an IVR effect when performing a task such as TUG locomotor task, with a young population which is at the peak of their functional abilities (this effect had been assessed in an elderly population affected by aging process). In our line of research, the next studies will precisely assess the IVR effect on more complex locomotor tasks, including interactions with the environment.

- Point 10:

Reviewer’s comments: 

Materials & Methods: Was there any ethical procedure or approval for this study? Did the authors provide informed consent to participate?

Authors response:

Our study was approved by an ethical committee: CPP EST-III, 10/12/2018 ; reference: N°ID-RCB: 2018-A02637-48. Participants gave an oral consent. 

- Point 11:

Reviewer’s comments: 

Materials & Methods: I believe the subsection ‘virtual reality device’ describes more than only the device. I suggest renaming the subsection with a different title.

Authors response:

Subsection title has been changed: “Immersive Virtual Reality task set up”

- Point 12:

Reviewer’s comments: 

Materials & Methods: Did the participants embody an avatar that walks in the VR scenario? What happens when participants wearing the HMD look down towards their own body? Did they see a virtual body?

Authors response:

There was no embodiment over an avatar. Participants evolved in the environment with first person view and had no visual feedback of their own body. When they looked down, they only saw the floor of the train, not their body. That has been clarified in the methodology section.

- Point 13:

Reviewer’s comments: 

Materials & Methods: The manuscript needs a more expanded description and explanation of the VR scenario. For instance, the following.

• In the VRm condition, when participants raise up from the chair, did participants walk in the same direction that the train was moving? Or did they walk in the direction from which the train is coming? This is an important consideration because we have evidence that optical flow affects behavior and motor performance.

• Also, when the participants were coming back to the chair, were they able to see the chair? Did the dimensions of the chair change according to the approximation of the participant?

• How was it guarantee that participants walked the same distance (3m) and turn around at the same spot in the normal and in the VR conditions? In the VR conditions, did participants need to overpass the suitcase or just get close to it?

• If possible, add a descriptive or explanatory video to better understand the performance of the task in VR.

Authors response:

• Optic flow is in the same direction as the subject during Go phase and in the opposite direction during Return phase. We clarified this point in methodology.

• Virtual chair was placed in the exact position of the real chair and was visible in the virtual environment. The dimensions of the chair still unchanged for each participant. I hope well understood your question.

• We systematically placed the real marker on the ground as well as the chair according to the virtual scene, distances were thus preserved between the conditions. Participants had to turn-around just in front of the suitcase rather than around, to allow an "on-spot" turn-around. It made possible to differentiate "pivot" turn-around from turn around with steps adjustments.

• Unfortunately, no explanatory video is available. 

- Point 14:

Reviewer’s comments: 

Materials & Methods: In the description of the Turn Around (TA) phase, the authors state “(TA) which starts when the subject begins to initiate the rotation, when actually trajectory changes to return to the chair” But these two events are not the same. One thing is “to initiate rotation” and a different thing is “trajectory changes to return to the char”. Can the authors specify where the analysis was made?

Authors response:

Thank you for the remark, there was an error in the description of this phase. The mistake has been revised.

- Point 15:

Reviewer’s comments: 

Materials & Methods: I would like to have a rationale and justification on why the authors chose time and number of steps as variables to be measured.

Authors response:

We measured the same indicators (time and number of steps) as in previous studies of our research team to be able to compare. More precisely, "time" is the main evaluation indicator to highlight a modification in performing the task; and number of steps represents adjustments of locomotion (for example: adopting a safe strategy lead to increase the number of steps, by reducing the length). Finally, time and number of steps are the primary indicators defining locomotion. 

Moreover, the placement of the camera, constrained by the space of the room, only allowed us to measure time and the number of steps reliably and accurately.

- Point 16:

Reviewer’s comments: 

Materials & Methods: Did the authors assess differences between the Go and Re phases within the VR environments? I would be curious to know the comparison results, if walking in a moving train, within a VR scenario, changes when one walks in the same direction than the train vs. a direction against the train.

Authors response:

It would have been very interesting but walking sections are very short and are dependent on the consecutive phases before and after, which can bias results. It would be necessary to have a longer distance. 

I still analyzed to give you an answer, there is no significant difference between "Go VRm" and "Go VR" on time and number of steps (when the optical flow is in the same direction). For the return phase, there is no difference in time but a difference in the number of steps, with an increase in "Return VRm" compared to "Return VR" when the optical flow scrolls in the opposite direction.

Results cannot be published for the limitations mentioned above.

- Point 17:

Reviewer’s comments: 

Materials & Methods: How was the variable “time” extracted? With a stopwatch in place during the experiment and further annotation? Or by analyzing the video in a post-experiment analysis?

Authors response:

Manual post-experiment video analysis, it has been mentioned in methodology section. 

- Point 18:

Reviewer’s comments: 

Materials & Methods: Please describe the equipment used to video record the participants.

Authors response:

We used a GoPro 1080p 60fps.

- Point 19:

Reviewer’s comments: 

Materials & Methods: Figure 1 is not cited in the manuscript. Also, a legend is missing. Again, I believe a video would be a best option to facilitate a better understanding on how the task is performed.

Authors response:

Figure 1 is now cited in the methodology, in “immersive virtual reality task set up” section. 

- Point 20:

Reviewer’s comments: 

Results: In Table 1, Why are there steps in the Turn Around and Sit-Down phases? Do we step when we sit down? How is it quantified?

Authors response:

There are different strategies to perform these phases: either need a pivot (a single step) or need multiple steps to rotate. We also considered adjustments before sitting down as steps.

Medians seemed similar from each condition, but variability (represented by IQR) showed a strongly increase in virtual reality conditions, that's why an analyze was needed. We have already measured significant differences in previous studies with elderly.

- Point 21:

Reviewer’s comments: 

Results: I am hesitant about the statistical presentation of the data. The authors say values from the 3 trials were averaged. Then in Table 1, authors use Medians. Is it the median of the mean values?

Authors response:

You perfectly understood. 

- Point 22:

Reviewer’s comments: 

Results: Also, for steps I understand the authors recorded the number of completed steps; this is, integer numbers such as 1, 2, 3, 4 or 5. Still we have data of steps such as 2,33 and 3,83. Another thing that is unclear: why the Total number is not the sum of the steps? Perhaps a better description of how data is handled will avoid confusion in the reader and serve for clarification.

Authors response:

We counted the number of steps for each phase. The total number of steps is equal to the sum of the steps recorded for each phase. 

Having 2.33 steps corresponds to the average of the 3 trials in a condition (for example: 2 trials with 2 steps and 1 trial with 3 steps for a condition, the average is therefore 2.33). 

The median of the total number of steps is different from the sum of the medians of each phase. That has been clarified in methodology.

- Point 23:

Reviewer’s comments: 

Results: Please provide better legends for all figures. Figures, together with legends, should be self-explanatory.

Authors response:

Title and legends have been improved. 

- Point 24:

Reviewer’s comments: 

Results: The Data presented in the Supplementary Material is the average from the 3 trials, correct? Can the authors provide the data of the 3 individual trials as well?

Authors response:

The data have been now provided. 

- Point 25:

Reviewer’s comments: 

Discussion: What message do the authors want to pass with the following sentence: “have adapted and modified their motor behavior during each phase, regardless of its complexity and sensory solicitation”? 

Authors response:

IVR seems to influence the variables measured regardless of the complexity of the phase concerned. The "time" and the "number of steps" increased in each phase, whether it was simple task such as go and return phases, or more complex task such as turn-around and sit-down.

- Point 26:

Reviewer’s comments: 

Discussion: How exactly can we get to the conclusion that: “It means that young adults have adapted and modified their motor behavior during each phase, regardless of its complexity and sensory solicitation”. I do not understand why some discussion sections revolve around adaptation. Did the authors assess adaptation?

Authors response:

The term motor adaptation has been removed. The term motor modification is more appropriate.

- Point 27:

Reviewer’s comments: 

Discussion: The discussion in general is lacking flow and structure. A comparison with the existing literature is seriously lacking, in particular with studies that have evaluated gait and motor changes during VR tasks.

Authors response:

Thank you very much for your comments, we improved discussion accordingly. 

- Point 28:

Reviewer’s comments: 

Discussion: The following sentence “Young adults could not see their own body” related to a previous comment of mine about avatar embodiment. Did the authors experience embodiment of an avatar to perform the task? Were participants assigned a virtual body?

Authors response:

There was no embodiment over an avatar. Participants evolved in the environment with first person view and had no visual feedback of their own body since they had no virtual body assigned. That has been clarified in the methodology section.

- Point 29:

Reviewer’s comments: 

Discussion: The authors state that “As participants have never had an immersive virtual reality experience before”. Was this an inclusion criterion during participant recruitment? How the authors assessed this?

Authors response:

Yes, it was an inclusion criterion added to the methodology section. That would allow another study to compare the difference with young people who already had virtual reality experiences. It would allow to quantify an eventual learning effect.

- Point 30:

Reviewer’s comments: 

Discussion: In general, I invite the authors to rethink the Discussion section, including incorporating the comments presented in the beginning of my review. In addition, a ‘study limitation’ section is missing. The authors may want to discuss what as well are the implications of the present study (e.g., scientific, clinical, technical).

Authors response:

A study limitation and perspective section has been added. 

- Point 31:

Reviewer’s comments: 

Discussion: The following information needs to appear in the Methods section: “In “VRm” condition, the virtual train is moving, with sounds and landscape scrolling outside the train. There is therefore a linear optical flow with constant speed, in the same direction as the subject during Go phase and in the opposite direction during Return phase.”

Authors response:

We applied your advice, thank you. 

- Point 32:

Reviewer’s comments: 

Discussion: This is good to have a reference such as 28, but others are lacking, and the discussion about the effect of optic flow should be expanded. Also, it should be incorporated in the introduction. I also invite the authors to consider reporting on an extra analysis that compares the Go and Return phases (since optic flow is different in these phases).

Authors response:

The Go and Return phases are unfortunately too short to analyze as an extra analysis. Moreover, these phases are dependent on the following phases.

- Point 33:

Reviewer’s comments: 

Discussion: In my opinion, the last 2 paragraphs of the discussion are important. First, I think they should appear as main topic of discussion. Second, references are missing. It is important that the authors review the existing literature and enrich this discussion. Below, I add a few of relevant studies.

Authors response:

We applied your advice, thank you. 

- Point 34:

Reviewer’s comments: 

Conclusion: Be objective in the conclusion, stick to the experiment results, and down tone overreaching claims.

Authors response:

We applied your advice, thank you. 

- Point 35:

Reviewer’s comments: 

General: There seems to be a lack of standardization on how the references are presented. Please review the reference format recommended by the journal.

Authors response:

We applied your advice, thank you.

---

## [Decision Letter · Decision Letter 1]

19 Aug 2022

PONE-D-22-05481R1Effects of using immersive virtual reality on time and steps during a locomotor task in young adultsPLOS ONE

Dear Dr. Renaux,

Thank you for submitting your manuscript to PLOS ONE. After careful consideration, we feel that it has merit but does not fully meet PLOS ONE’s publication criteria as it currently stands. Therefore, we invite you to submit a revised version of the manuscript that addresses the points raised during the review process.

We look forward to receiving your revised manuscript.

Kind regards,

Imre Cikajlo, Ph.D.

Academic Editor

PLOS ONE

\\

Additional Editor Comments (if provided):

Please read carefully the remarks and issues raised by the reviewers and revise the manuscript. Focus on the major issues that may present the valuable information considerable for publication of the manuscript.

Reviewers' comments:

Reviewer's Responses to Questions

**Comments to the Author**

1. If the authors have adequately addressed your comments raised in a previous round of review and you feel that this manuscript is now acceptable for publication, you may indicate that here to bypass the “Comments to the Author” section, enter your conflict of interest statement in the “Confidential to Editor” section, and submit your "Accept" recommendation.

Reviewer #1: (No Response)

Reviewer #2: (No Response)

2. Is the manuscript technically sound, and do the data support the conclusions?

Reviewer #1: Yes

Reviewer #2: Partly

3. Has the statistical analysis been performed appropriately and rigorously? 

Reviewer #1: Yes

Reviewer #2: Yes

4. Have the authors made all data underlying the findings in their manuscript fully available?

Reviewer #1: Yes

Reviewer #2: Yes

5. Is the manuscript presented in an intelligible fashion and written in standard English?

Reviewer #1: Yes

Reviewer #2: Yes

6. Review Comments to the Author

Reviewer #1: I would like to first thanks authors to consider my comments.

The paper has improved. Authors added several relevant references related to their work. The rationale of the study is now better justified and discussed with respect to previous studies. Some methodological details have also been added which help to better understand the study.

I still have some minor comments detailed below, that could be fixed easily:

• Regarding the “as fast as possible speed” vs “comfort speed” to perform the task, I do not think this is the same idea as proposed in authors’ answer. Some recent work considered the effect of the effect of the instruction provided to participants in a TUG task and show different timing. E.g. : https://www.xsens.com/cases/normative-data-set-timed-go-component-times-different-conditions.

• Answer to point 8: I agree that the reduced field of view may have provoked different coordination strategies regarding head motion. This is for example shown in the following paper where gaze and head motion anticipate more the change of direction in VR than in real conditions. Brument, H., Podkosova, I., Kaufmann, H., Olivier, A. H., & Argelaguet, F. (2019, March). Virtual vs. physical navigation in vr: Study of gaze and body segments temporal reorientation behaviour. In 2019 IEEE Conference on Virtual Reality and 3D User Interfaces (VR) (pp. 680-689). But I would not talk about the naturalness of the motion. Participants may have adapted to this new perceptual condition. I would suggest to remove “making the movement less natural” line 303.

• Lines 94-95: to strengthen authors’ contribution, I would suggest to add some words about the specificity of their study => for better understanding IVR effects during a functional and contextualized task,

• Lines 214: can authors provide some details about which criterion was used to determine the initiation of the turn? Even if it was done manually, that is of interest to know if the experimenters based their analysis on head motion (which starts reorientation before the other segment, and sooner in VR than in VE) or another body part.

• Study limitation, line 346: I do not think authors should talk about the short distance due to the constraint of IVR. Indeed, the TUG task is based on a 3m straight walking path, so it makes sense to have chosen this distance. Also, with 4 base stations, it is almost possible to capture a 10m walking path.

• Line 354: it would be interesting that authors provide some examples of the kinematic and kinetic determinants that can be considered for future work.

Reviewer #2: The authors have improved the manuscript. For instance, by including limitations of the task, the sensorimotor constraints imposed by wearing a VR HMD, expanding the comparison to existing literature, reorganizing generally the manuscript, and contextualizing the line of studies in the research group. Nevertheless, there are still aspects of the manuscript that demand revisions and/or further clarifications.

First, I would like to ask the authors that, in case of follow up reviews, please specify when the answers (to my comments) are accompanied by actual changes in the manuscript. When this is the case, please mention in your response to the comment the lines and/or pages of the edited content in the revised manuscript. As the authors will see in this review, many of my comments ask simply to incorporate your answers to the comments of my preview review into the manuscript (since in many cases it is not clear whether the authors only reply to me or do actually something in the manuscript).

The authors state that “this study is part of a line of studies”, and they based on this fact their justification for the selection of TUG and of the chosen variables. This is good to know and more importantly, it should be clearly reflected in the manuscript. At the same time, this cannot be the only justification. The manuscript needs to better explain and contextualize the relation of the present study to the larger “line of studies” the authors are conducting, including the previous ones (already published) and the planned ones, and stating what are the main research questions and objectives of such “lines of studies”.

Can you please provide me with the reference concluding that “time and number of steps are the primary indicators defining locomotion”?

The answer of the authors to my comment regarding turn around and sit-down phases is not satisfactory. Again, I believe steps is not an optimal measure, especially for these phases. In the original Table 1, the 3 values for turn around were 2,00 (0,08), 2,33 (1,00) and 2,00 (1,00); and the 3 values for sit down were 2,00 (0,33), 2,00 (0,67) and 2,33 (0,67). These are comparable numbers among conditions. Also, authors mention that young adults can turn-around with a pivot in real condition. So, perhaps a description of how the analysis was performed for these 2 phases (e.g., of this pivot) is important?

The authors’ answer to my comment does not demonstrate that walking in a virtual environment can be done “unconsciously” and “without additional cognitive effort, as in real life”. The 2 sentences provided by the authors do not mention neither consciousness nor cognition. The sentence by Fuchs, 2018, that the authors provide into their response, may suggest that humans assess the contextual environment and take behavioral decisions in VR following a method like the one we use to assess environments and take decisions in real life; however, this cannot be extrapolated to infer walking occurs unconsciously or without additional cognitive effort in VR. Healthy young individuals such as the sample in this study can walk (in real life) without much cognitive deliberation, that is true. Overall, walking may indeed be performed largely automatically: an automated behavior that requires little to no attention. However, automaticity is not unconsciously. Also, cognition is a complex concept. If the authors are convinced that walking in a VR environment has no additional cognitive effort, I prefer seeing the citation of a study that measured cognitive effort and/or provide sufficient proof to conclude that.

In their response to one of my comments, the authors state that “This study makes possible to confirm that it does exist an IVR effect when performing a task such as TUG locomotor task, with a young population which is at the peak of their functional abilities”. But the revised manuscript (rightly) discusses that the inexperience of participants with VR may be a co-factor causing the observed IVR effect. I would assume that after training and motor learning, healthy young adults would engage in a process of adaptation and incorporate automatisms that will either decrease or remove any IVR effect in the TUG task. If my assumption is correct, the reason of the findings in this study may be the lack of experience with a VR headset, and not VR itself.

This is a pity that the authors have no explanatory video available. Still, concerning the virtual chair and the virtual suitcase, how were their dimensions fixed, was it in proportion to other objects in the virtual scenario? There was a perspective effect? For instance, did the dimensions of the virtual chair and suitcase increase while participants approach? I would like to understand this more from a 3D design and programming point of view, i.e., how was this dealt with in Unity or Unreal?

The manuscript does not specify which gaming engine was used to generate the VR environment.

Please incorporate your response to “Point 15” in the manuscript (ideally the response to each comment should come with an indication of where the change was made, e.g., subsection, number of pages, line number. Otherwise, it makes difficult to know if the authors took any action to address the reviewer comments).

Concerning the sub-analysis conducted in “point 16”. The authors say that walking sections are very short in distance and dependent to other TUG phases. Nevertheless, the study main results are based on the comparison of these walking sessions among conditions…. How to explain that such limitations prevent the publication of only one type of comparison, but not of other comparisons?

Following up on point 20. The authors write that “there are different strategies to perform these phases”. So please, clearly mention those strategies in the manuscript, and specify how strategies to turn around and sit-down change between young healthy and elderly. Also, since no answer was given, I repeat the question: how the authors using video records calculate number of steps during these two phases? E.g., what qualifies as a step when we sit down?

Make sure comments and responses in point 21 are incorporated in the manuscript.

One answer to one of my comments says that no experience in VR was an inclusion criterion, and that it would allow another study to compare the difference with young people who already had VR experiences. The manuscript ideally should include this. It reinforces again how a rationale and contextualization based on the studies of the group in this line of research would facilitate justifications for the experimental design, like here for this inclusion criteria.

I repeat one comment that had no answer: This is good to have a reference such as 28, but others are lacking, and the discussion about the effect of optic flow should be expanded.

I suggest going throughout the paper again for improving minor English mistakes and words cohesion within and among sentences. Also, there is information that appears multiple times in the text. My suggestion is to avoid repeating the same information. It will make the manuscript shorter, clearer, and easier to read.

In several instances of the manuscript (this comment related to another comment above), the authors reinforce something in the lines “Finally, “time” and “number of steps” are the primary global indicators defining locomotion”. Is this an opinion of the authors or do we actually have a comprehensive study that conclude this? Or were the variables chosen due to methodological constraints? In point 21, it suggests that the reason to choose these variables were “the placement of the camera, constrained by the space of the room, only allowed us to measure time and the number of steps reliably and accurately”. Please clarify.

The authors write: “The visual exproprioceptive feedback of body segments during movement is normally used, even unconsciously [33]”. Again, the word (un-)consciously appears in the manuscript, and I could not see how the manuscript by Patla, 1998, can be a reference for such a statement. I ask the authors to explain. Furthermore, I invite them to be careful when rephrasing other studies or when using other studies as a way to consolidate an idea.

In the last paragraph of the introduction, just before specifying the aim of the study, the authors mention the possible expected results of this study and the hypothesis for comparing with the previous study with elderly (lines 118-123): “If aging process alone justifies motor changes measured in our previous studies…”. I appreciate that the authors included this. However, I think that it needs to be revisited in the discussion as well, for instance providing a closing-loop comment where the authors compare the results of both studies.

7. PLOS authors have the option to publish the peer review history of their article (what does this mean?). If published, this will include your full peer review and any attached files.

Reviewer #1: No

Reviewer #2: **Yes: **Desiderio Cano Porras

---

## [Author Response · Author response to Decision Letter 1]

15 Sep 2022

Response to Reviewer #1

- Point 1: 

Reviewer’s comments: 

Regarding the “as fast as possible speed” vs “comfort speed” to perform the task, I do not think this is the same idea as proposed in authors’ answer. Some recent work considered the effect of the instruction provided to participants in a TUG task and show different timing. E.g. : https://www.xsens.com/cases/normative-data-set-timed-go-component-times-different-conditions.

Authors response:

The study you mentioned effectively shows that results of the TUG test may differ according to the kind of instruction given. However, in our studies, the exact same instruction was given to the elderly and young adults: The oral instruction was “You must perform the task by walking as quickly as possible without running or putting yourself in danger”. This methodology part has been modified, the given instruction has been written. 

Finally, Podsiadlo and Richardson (1991) evoke the notion of comfortable speed to describe the test. In young people, the dangerous aspect of a rush that can lead to an imbalance (risk of falling) is negligible compared to the temptation of running, which could create a bias for the test. 

Incorporated in the manuscript lines 147-148.

- Point 2:

Reviewer’s comments: 

Answer to point 8: I agree that the reduced field of view may have provoked different coordination strategies regarding head motion. This is for example shown in the following paper where gaze and head motion anticipate more the change of direction in VR than in real conditions. Brument, H., Podkosova, I., Kaufmann, H., Olivier, A. H., & Argelaguet, F. (2019, March). Virtual vs. physical navigation in vr: Study of gaze and body segments temporal reorientation behaviour. In 2019 IEEE Conference on Virtual Reality and 3D User Interfaces (VR) (pp. 680-689). But I would not talk about the naturalness of the motion. Participants may have adapted to this new perceptual condition. I would suggest to remove “making the movement less natural” line 303.

Authors response:

The term "natural" is not relevant and has been removed. Indeed, it is a more anticipated and amplified head movement in VR than in Real (in agreement with the study you proposed). It actually induces an adaptation of the head motion. Thank you for your reference study.

Incorporated in the manuscript lines 331-332.

- Point 3:

Reviewer’s comments: 

Lines 94-95: to strengthen authors’ contribution, I would suggest to add some words about the specificity of their study => for better understanding IVR effects during a functional and contextualized task,

Authors response:

We applied your advice, thank you. 

Incorporated in the manuscript line 118. 

- Point 4:

Reviewer’s comments: 

Lines 214: can authors provide some details about which criterion was used to determine the initiation of the turn? Even if it was done manually, that is of interest to know if the experimenters based their analysis on head motion (which starts reorientation before the other segment, and sooner in VR than in VE) or another body part.

Authors response:

The criterion used for determining the turn-around start was the moment when the swing foot overlaps the stance foot, before contacting the ground, pivoted to initiate the rotation. It makes the steps count easier with a cut during overlapping. Time and number of steps cuts were always done between two steps.

The entire phase cut method has been added in the method section (the start of every phase determines the end of the previous phase).

- The start of Get Up phase: when the back takes off from the chair.

- The start of Go phase: when a foot takes off to initiate the first step

- The start of Turn-Around phase: when the swing foot overlaps the stance foot, pivoted to initiate the rotation before contacting the ground.

- The start of Return phase: when the swing foot overlaps the stance foot, pivoted in direction of the chair before contacting the ground. 

- The start of Sit-Down phase: when the swing foot overlaps the stance foot, pivoted to initiate the rotation before sitting down on the chair. 

- The end of the task: when the back has its initial position, back leaning against the backrest.

Incorporated in the manuscript lines 199 to 216.

- Point 5:

Reviewer’s comments: 

Study limitation, line 346: I do not think authors should talk about the short distance due to the constraint of IVR. Indeed, the TUG task is based on a 3m straight walking path, so it makes sense to have chosen this distance. Also, with 4 base stations, it is almost possible to capture a 10m walking path.

Authors response:

This comment referred only to the “walking” phases studied. It would be interesting to study the effect on walking over a longer distance. The term gait was more appropriate, but this part has been removed in agreement with your relevant comments. Others more extensive tracking systems exist, with 4 base stations or motion capture, it would be possible to track gait (unlike using treadmill).

- Point 6:

Reviewer’s comments: 

Line 354: it would be interesting that authors provide some examples of the kinematic and kinetic determinants that can be considered for future work.

Authors response:

In future studies currently being conducted, whole body kinematics data (position, velocity, and acceleration of each body segment) will be analysed when performing complex motor tasks with interactions in the virtual environment. Locomotor and postural data will then be measured. A kinetic analysis of the variation of the centre of pressure will be carried out.

Incorporated in the manuscript lines 396 to 398. 

Response to Reviewer #2

- Point 1: 

Reviewer’s comments: 

The authors state that “this study is part of a line of studies”, and they based on this fact their justification for the selection of TUG and of the chosen variables. This is good to know and more importantly, it should be clearly reflected in the manuscript. At the same time, this cannot be the only justification. The manuscript needs to better explain and contextualize the relation of the present study to the larger “line of studies” the authors are conducting, including the previous ones (already published) and the planned ones, and stating what are the main research questions and objectives of such “lines of studies”.

Authors response:

The main objective of the line of research is to investigate whether immersive virtual reality (IVR) could be used as a new tool for assessing an individual's functional abilities and risk of falling. Indeed, IVR makes possible to produce, contextualize and simulate an environment of daily life. This tool might be deployed in different places since it's digital and potentially highly reproductible wherever.

Assessing individual's functional capacities in an IVR environment has the advantages being standardized (choice of scenarios and proposed stimulations), modifiable and adaptable (parameterization of the proposed stimulations = triggering, speed, trajectory of a mobile obstacle for instance) and secure (experimenter is always present but invisible from the point of view of participant, so he is not influenced when carrying out the task. In the case of an obstacle clearance that could cause tripping, there is no physical danger of tripping over virtual obstacle, moreover the participant does not have to trip on the obstacle for us to know if there was any collision, and thus a falling risk).

Many preliminary studies are necessary to reach this objective. First, we wanted to know if it does exist an IVR effect when performing the same task in real and in IVR conditions. We therefore evaluated the consequences of immersive virtual reality on primary motor control indicators when performing a well-known locomotor task (in the "Timed Up and Go" test) in elderly people. The results showed that there was an increase in the number of steps and time to complete the same task in VR compared to real condition. It means a decrease in performance in IVR condition. Following it, we wanted to determine if it was due to the technology or the effect of aging process, or a combination of both. We therefore reproduced this same experiment with a population of young adults (current study).

The results with young adults are similar, with an increase in the number of steps and time. It means it does exist an IVR effect regardless of the population studied. We will then determine whether the magnitude of this effect is equivalent according to age. We now know that performing the same task in IVR compared to real involves a decrease in motor performance.

The next steps consist in enriching the proposed tasks with interactions in the IVR environment to assess the functional abilities of individuals. Kinematic and kinetic sensors will then be used to analyse the motor skills of participants more finely. Once again, we will compare the results between elderly and young adults to better understand the adaptations involved.

Finally, the last step will consist in correlating the fall history of the individuals over the months following the experiment with their performance in IVR and to create a learning machine which might be used as new method for a fall risk assessment. 

Incorporated on the whole manuscript (examples: lines 67 to 72; 393 to 399). 

- Point 2: 

Reviewer’s comments: 

Can you please provide me with the reference concluding that “time and number of steps are the primary indicators defining locomotion”?

Authors response:

The sentence must be reworded in a more relevant way “time and number of steps are primary indicators of locomotion”

Based on study from Hollman, J. H., McDade, E. M., & Petersen, R. C. (2011). Normative spatiotemporal gait parameters in older adults. Gait & posture, 34(1), 111-118., There are several primary domains of spatiotemporal gait performance identified. The one that interests us is the “pace” domain characterized by parameters that included gait speed, step length and stride length. In our studies, considering the placement constraint of our camera, the step length and stride length could not be measured reliably, that is the reason why we just measured the number of steps and the time taken for each phase. With a fixed distance (here 3 meters), the walking speed depends on time, the average step length depends on number of steps. Concerning Turn-Around and Sit-Down phases, it was interesting to keep the same variables to be consistent with the other phases and to reflect motor strategies employed.

Incorporated in the manuscript lines 191 to 197. 

- Point 3: 

Reviewer’s comments: 

The answer of the authors to my comment regarding turn around and sit-down phases is not satisfactory. Again, I believe steps is not an optimal measure, especially for these phases. In the original Table 1, the 3 values for turn around were 2,00 (0,08), 2,33 (1,00) and 2,00 (1,00); and the 3 values for sit down were 2,00 (0,33), 2,00 (0,67) and 2,33 (0,67). These are comparable numbers among conditions. Also, authors mention that young adults can turn-around with a pivot in real condition. So, perhaps a description of how the analysis was performed for these 2 phases (e.g., of this pivot) is important?

Authors response:

The most important result in this analysis corresponds to the variability (expressed by the interquartile range). There is little variability in Turn Around in Real (0.08), but a high variability in Turn Around in VR / VRm conditions (1.00). It means that a part of young adults has modified their strategy for performing the Turn Around in VR conditions compared to real condition. 

The phase cut method has been clarified to the method section for better understanding. Each counted step is a weight transfer from one foot to the other one, even if trampling. 

The pivot strategy consists of transferring the body weight to the stance foot to rotate the trunk at 180° and to redirect the swing foot towards the return phase. This strategy requires 2 steps (one step to pivot, and one step before starting return phase). One participant performed a pivot by sliding their foot on the floor during the body rotation, and therefore only used one step. The multi-step strategy consists of performing this same 180° body rotation by using several steps (at least 3).

Incorporated in the manuscript lines 199 to 216 and lines 280 to 294. 

- Point 4: 

Reviewer’s comments: 

The authors’ answer to my comment does not demonstrate that walking in a virtual environment can be done “unconsciously” and “without additional cognitive effort, as in real life”. The 2 sentences provided by the authors do not mention neither consciousness nor cognition. The sentence by Fuchs, 2018, that the authors provide into their response, may suggest that humans assess the contextual environment and take behavioral decisions in VR following a method like the one we use to assess environments and take decisions in real life; however, this cannot be extrapolated to infer walking occurs unconsciously or without additional cognitive effort in VR. Healthy young individuals such as the sample in this study can walk (in real life) without much cognitive deliberation, that is true. Overall, walking may indeed be performed largely automatically: an automated behavior that requires little to no attention. However, automaticity is not unconsciously. Also, cognition is a complex concept. If the authors are convinced that walking in a VR environment has no additional cognitive effort, I prefer seeing the citation of a study that measured cognitive effort and/or provide sufficient proof to conclude that.

Authors response:

We meant what you wrote “humans assess the contextual environment and take behavioral decisions in VR following a method like the one we use to assess environments and take decisions in real life”, this is the interpretation we have done from Fuchs. Therefore, you are totally right about the nuance between "unconscious" and "automaticity", that is why we have changed the formulation of our sentences.

Modified in the manuscript line 58.

- Point 5: 

Reviewer’s comments: 

In their response to one of my comments, the authors state that “This study makes possible to confirm that it does exist an IVR effect when performing a task such as TUG locomotor task, with a young population which is at the peak of their functional abilities”. But the revised manuscript (rightly) discusses that the inexperience of participants with VR may be a co-factor causing the observed IVR effect. I would assume that after training and motor learning, healthy young adults would engage in a process of adaptation and incorporate automatisms that will either decrease or remove any IVR effect in the TUG task. If my assumption is correct, the reason of the findings in this study may be the lack of experience with a VR headset, and not VR itself.

Authors response:

Indeed, there is likely a process of adaptation when the task is repeated several times, with consequence a decrease in the IVR effect. Participants would get accustomed with the HMD and the IVR environment.

Nevertheless, the aim is to evaluate the IVR effect during a first VR experience. The main objective of this line of research is to figure out whether this technology could be used as a tool for assessing functional abilities or risk of falling. Thus, people who will be subjected to this test will not necessarily have the opportunity to practice IVR before. In addition, elderly sample who carried out this same protocol had never experienced IVR.

However, it may be relevant to specify that it is an IVR effect after a first IVR experience. Beyond a probable learning effect, it may be a familiarization effect, a greater ease.

Perspectives: after living several IVR experiences, it would be interesting to measure the learning effect at several temporalities. It will be done in a future study.

Incorporated in the manuscript lines 24, 268, and 317 to 325. 

- Point 6: 

Reviewer’s comments: 

This is a pity that the authors have no explanatory video available. Still, concerning the virtual chair and the virtual suitcase, how were their dimensions fixed, was it in proportion to other objects in the virtual scenario? There was a perspective effect? For instance, did the dimensions of the virtual chair and suitcase increase while participants approach? I would like to understand this more from a 3D design and programming point of view, i.e., how was this dealt with in Unity or Unreal?

Authors response:

All the proportions of the objects were consistent with a real environment, everything was on a human scale. There was a perspective effect equivalent to the perspective effect in real. For instance, the dimensions of the chair and the suitcase changed according to the approach or the distance from the participant.

The development was done on Unity by internship students. They had specifications, but we don't know more about "trade secrets".

Incorporated in the manuscript line 164 to 166. 

- Point 7: 

Reviewer’s comments: 

The manuscript does not specify which gaming engine was used to generate the VR environment.

Authors response:

The virtual application was developed on Unity. 

Incorporated in the manuscript line 159. 

- Point 8: 

Reviewer’s comments: 

Please incorporate your response to “Point 15” in the manuscript (ideally the response to each comment should come with an indication of where the change was made, e.g., subsection, number of pages, line number. Otherwise, it makes difficult to know if the authors took any action to address the reviewer comments).

Authors response:

It’s now incorporated in the manuscript.

Incorporated in the manuscript lines 190 to 198. 

- Point 9:

Reviewer’s comments: 

Concerning the sub-analysis conducted in “point 16”. The authors say that walking sections are very short in distance and dependent to other TUG phases. Nevertheless, the study main results are based on the comparison of these walking sessions among conditions…. How to explain that such limitations prevent the publication of only one type of comparison, but not of other comparisons?

Authors response:

We do not want to compare "Go" VRm and "Return" VRm because the consecutive phases before and after are different and the influence of this factor on three or four steps could bias the results and not showing an effect of the optic flow. The exact same task would have to be performed for assessing an effect of optic flow direction.

- Point 10: 

Reviewer’s comments: 

Following up on point 20. The authors write that “there are different strategies to perform these phases”. So please, clearly mention those strategies in the manuscript, and specify how strategies to turn around and sit-down change between young healthy and elderly. Also, since no answer was given, I repeat the question: how the authors using video records calculate number of steps during these two phases? E.g., what qualifies as a step when we sit down?

Authors response:

There are two main strategies to perform the Turn-Around (described point 3): The pivot strategy consists of transferring the body weight to the stance foot to rotate the trunk at 180° and to redirect the swing foot towards the return phase. This strategy requires 2 steps (one step to pivot, and one step before starting return phase). One participant performed a pivot by sliding their foot on the floor during the body rotation, and therefore only used one step. The multiple step strategy consists of performing this same 180° body rotation by using several steps (at least 3). Young adults’ resort to these two strategies whereas elderly always need multiple steps. Indeed, elderly have a safer motor strategy, with a wider sustentation polygon, a slower body rotation, a greater gathering of visual information by using fixed landmarks in the environment to stabilize their balance. Overall, they are less dynamic for reducing the risk of imbalance.

The strategies are now mentioned in the manuscript lines 280 to 290.

The steps counted during the "Sit Down" phase correspond to the one used during the turn-around before sitting on the chair. Two main possible strategies to perform Sit Down: either complete turn-around in an erected posture then sitting (adjustment of the position related to the chair to feel the chair behind + hands contacting with the chair to ensure controlled sitting in the right place and compensating a muscle deficit), or a turn-around while lowering the center of gravity and starting sitting action. Because of these two strategies, it was in some cases impossible to dissociate the turn-around from the sit.

The method for calculating the number of steps is clarified in the method section lines 199 to 216.

- Point 11:

Reviewer’s comments: 

Make sure comments and responses in point 21 are incorporated in the manuscript.

Authors response:

It’s now incorporated in the manuscript lines 220.

- Point 12:

Reviewer’s comments: 

One answer to one of my comments says that no experience in VR was an inclusion criterion, and that it would allow another study to compare the difference with young people who already had VR experiences. The manuscript ideally should include this. It reinforces again how a rationale and contextualization based on the studies of the group in this line of research would facilitate justifications for the experimental design, like here for this inclusion criteria.

Authors response:

We justified this inclusion criterion more precisely in the discussion section. The reasons are partly mentioned in point 5. Your suggestion is an excellent perspective to be considered in a future study.

Incorporated in the manuscript lines 319 to 325.

- Point 13:

Reviewer’s comments: 

I repeat one comment that had no answer: This is good to have a reference such as 28, but others are lacking, and the discussion about the effect of optic flow should be expanded.

Authors response:

You mean reference number 40 instead of number 28? We tried to better explain the effect of optic flow with references 39, 40, 41, 42, 43 and 44. 

- Point 14:

Reviewer’s comments: 

I suggest going throughout the paper again for improving minor English mistakes and words cohesion within and among sentences. Also, there is information that appears multiple times in the text. My suggestion is to avoid repeating the same information. It will make the manuscript shorter, clearer, and easier to read.

Authors response:

We applied your advice, thank you. 

- Point 15:

Reviewer’s comments:

In several instances of the manuscript (this comment related to another comment above), the authors reinforce something in the lines “Finally, “time” and “number of steps” are the primary global indicators defining locomotion”. Is this an opinion of the authors or do we actually have a comprehensive study that conclude this? Or were the variables chosen due to methodological constraints? In point 21, it suggests that the reason to choose these variables were “the placement of the camera, constrained by the space of the room, only allowed us to measure time and the number of steps reliably and accurately”. Please clarify.

Authors response:

Time and number of steps are primary indicators of locomotion (linked to speed and length) but also, they are the only variables that can be measured accurately and reliably with the constraints related to the placement of the camera. Point 2 answers this question. 

Incorporated in the manuscript lines 190 to 198.

- Point 16:

Reviewer’s comments: 

The authors write: “The visual exproprioceptive feedback of body segments during movement is normally used, even unconsciously [33]”. Again, the word (un-)consciously appears in the manuscript, and I could not see how the manuscript by Patla, 1998, can be a reference for such a statement. I ask the authors to explain. Furthermore, I invite them to be careful when rephrasing other studies or when using other studies as a way to consolidate an idea.

Authors response:

The term “unsconsciously” is not relevant, it has been removed. 

- Point 17:

Reviewer’s comments: 

In the last paragraph of the introduction, just before specifying the aim of the study, the authors mention the possible expected results of this study and the hypothesis for comparing with the previous study with elderly (lines 118-123): “If aging process alone justifies motor changes measured in our previous studies…”. I appreciate that the authors included this. However, I think that it needs to be revisited in the discussion as well, for instance providing a closing-loop comment where the authors compare the results of both studies.

Authors response:

We applied your advice, thank you.

---

## [Decision Letter · Decision Letter 2]

26 Sep 2022

Effects of using immersive virtual reality on time and steps during a locomotor task in young adults

PONE-D-22-05481R2

Dear Dr. Renaux,

We’re pleased to inform you that your manuscript has been judged scientifically suitable for publication and will be formally accepted for publication once it meets all outstanding technical requirements.

Kind regards,

Imre Cikajlo, Ph.D.

Academic Editor

PLOS ONE

Additional Editor Comments (optional):

Reviewers' comments:

Reviewer's Responses to Questions

**Comments to the Author**

1. If the authors have adequately addressed your comments raised in a previous round of review and you feel that this manuscript is now acceptable for publication, you may indicate that here to bypass the “Comments to the Author” section, enter your conflict of interest statement in the “Confidential to Editor” section, and submit your "Accept" recommendation.

Reviewer #1: All comments have been addressed

Reviewer #2: All comments have been addressed

2. Is the manuscript technically sound, and do the data support the conclusions?

Reviewer #1: (No Response)

Reviewer #2: Yes

3. Has the statistical analysis been performed appropriately and rigorously? 

Reviewer #1: (No Response)

Reviewer #2: Yes

4. Have the authors made all data underlying the findings in their manuscript fully available?

Reviewer #1: (No Response)

Reviewer #2: Yes

5. Is the manuscript presented in an intelligible fashion and written in standard English?

Reviewer #1: (No Response)

Reviewer #2: Yes

6. Review Comments to the Author

Reviewer #1: I would like to thank authors for addressing my comments. I feel that this manuscript is now acceptable for publication

Reviewer #2: The authors have addressed my comments satisfactorily, including the specifications regarding the methods for analysis of the TUG evaluated phases, comparison to other studies, and expanding the discussion to cover constraints of the study design and results interpretation. By finalizing this review, I would like the authors to note two things. First, it could be argued that number of steps depends on step length, and not the other way around. Second, center of gravity was not measured, so statements about it demand being cautious.

7. PLOS authors have the option to publish the peer review history of their article (what does this mean?). If published, this will include your full peer review and any attached files.

Reviewer #1: No

Reviewer #2: **Yes: **Dr. Desiderio Cano Porras

---

## [Editor Report · Acceptance letter]

28 Sep 2022

PONE-D-22-05481R2 

Effects of using immersive virtual reality on time and steps during a locomotor task in young adults 

Dear Dr. Renaux:

I'm pleased to inform you that your manuscript has been deemed suitable for publication in PLOS ONE. Congratulations! Your manuscript is now with our production department. 

Kind regards, 

on behalf of

Professor Imre Cikajlo 

Academic Editor

PLOS ONE